# KDP: Simplifying Representation Dynamics in Kernel Space

**Zeyu Ma**[1,2]**, Wanying Wang**[2]**, Guchu Zou**[3]**, Mingang Chen**[2]**, Jianhong Wu**[1]*

[1]Shanghai Normal University
[2]Shanghai Key Laboratory of Computer Software Testing and Evaluating
[3]Shanghai Institute of Ceramics, Chinese Academy of Sciences
`wujianhong@shnu.edu.cn,`
`{mzy, wangwy, cmg}@sscenter.sh.cn,`
`zouguchu@mail.sic.ac.cn`

## Abstract

This paper proposes Kernelized Dynamics Pruning (KDP), a novel layer pruning method from the perspective of simplifying representation dynamics within large language models (LLMs). Motivated by the high similarity between consecutive layer representations, we view the LLM's forward pass as a discrete-time dynamical system. We speculate that this phenomenon indicates the model's internal dynamics have entered a "slow manifold", which exhibits computational redundancy. Based on this insight, we project the representations into a kernel space where the complex, non-linear transformation between them is simplified to an approximately linear one. Then, a simple network learns the inverse kernel transformation, thereby enabling the pruning of the entire layer block. Both theoretical analysis and extensive experiments validate the effectiveness of KDP, demonstrating its superiority over existing pruning baselines.

## 1 Introduction

Large language models (LLMs) have demonstrated exceptional performance across various tasks and domains (Touvron et al., 2023; Grattafiori et al., 2024; Yang et al., 2025; Liu et al., 2024). However, as models increase in scale, the substantial computational and hardware deployment costs pose significant challenges, limiting their broader application in real-world scenarios. Pruning is a primary technique for model compression aimed at alleviating the aforementioned problems (Louizos et al., 2017; Xia et al., 2023; Sun et al., 2024; Ma et al., 2023). Among various pruning methods, layer pruning is garnering increasing attention because it naturally leads to inference acceleration and model size reduction without requiring special handling.

The prevailing layer pruning paradigm involves removing redundant layers (Song et al., 2024; Men et al., 2024) or substituting them with compact modules (Yang et al., 2024; Chen et al., 2025), with a primary emphasis on the choice of pruning locations and the techniques for performance recovery (e.g., fine-tuning or distillation). However, this line of work largely overlooks the fundamental properties of the model's internal dynamic flow, thereby leaving the potential simplification patterns unexplored, see Appendix C for details on related work. We suggest that pruning can be viewed not only as a method to "construct" a smaller sub-network, but also as a process to "search" for a geometric viewpoint that reveals the inherent simplicity of complex dynamics.

LLMs exhibit a notable characteristic: their consecutive layers tend to learn highly similar representations (Ding et al., 2025). As demonstrated in Figure 1, we employ both the centered kernel alignment (CKA, Kornblith et al., 2019) and the cosine similarity metric to measure the similarity of layer representations from the perspectives of the original space and the kernel space, respectively, and find that multiple consecutive layers exhibit consistently high similarity. This observation raises a key research question: *Does representational similarity equate to computational redundancy which can be substituted by a simple function*?

---

*Corresponding author: Jianhong Wu

Viewing the forward pass of an LLM as a discrete-time dynamical system, high representational similarity signifies the system's entry into a "slow manifold", characterized by a small velocity. We assume that in such regions, the system's short-term evolution can be described by a much simpler function (e.g., a linearized first-order approximation), rather than requiring the nonlinear dynamics of a complete Transformer block. However, substituting the non-linear forward pass with linear dynamics may result in the loss of fine-grained details, which could contain low-variance, task-relevant information (Cloos et al., 2025). Notably, Figure 1 shows that the similarity measured in the kernel space is substantially higher than that in the original space. This observation suggests that the kernel space possesses stronger capability in modeling high-dimensional relationships, consistent with the intuition behind SVMs (Cortes & Vapnik, 1995). Consequently, it exhibits higher similarity and suggests a potential for linear simplification. Motivated by this, we aim to identify a Hilbert space in which the complex dynamics become amenable to linear approximation, thereby enabling effective layer pruning via linear simplification in kernel space.

Based on these insights, we propose **Kernelized Dynamics Pruning (KDP)**, simplifying representational dynamics within a kernel space to reduce computational redundancy. Specifically, our method proceeds in the following two steps. First, we jointly optimize a learnable kernel transformation and the linear coefficients mapping between layers. Second, an inverse transformation network maps these representations from the kernel space back to the original space, reconstructing the original layer to enable model pruning.

We validate the effectiveness of our method through both theoretical analysis and extensive experiments. First, we present a theorem providing an error bound for approximating multi-layer representations with linear transformations in the kernel space, thereby demonstrating that cross-layer representations can be linearly approximated within this space. We further show that the linear approximation capability in the kernel space surpasses that in the original space, providing a theoretical foundation for KDP. Extensive experimental results empirically answer our guiding research questions, demonstrating that exploiting representation similarity significantly simplifies computational processes by eliminating redundancy. KDP learns simplified internal representational dynamics in kernel space using only localized representation supervision, without requiring fine-tuning of the entire model on downstream tasks. This method enables effective modeling with limited data and obviates the need for post-training to restore performance. Our main contributions are as follows:

- We reformulate layer pruning as the search for an optimal geometric embedding within a Reproducing Kernel Hilbert Space and propose Kernelized Dynamics Pruning (KDP) method, which linearizes representational dynamics using the kernel trick to simplify consecutive layers.

- We provide a theoretical error bound for linearization in the kernel space and demonstrate the modeling and linear simplification advantages of the kernel space, with experimental results corroborating the theoretical analysis.

- We conduct extensive experiments on 15 benchmarks, demonstrating that our method maintains superior performance while requiring only a small number of trainable parameters and limited calibration data, eliminating the need for additional post-training.

## 2 Preliminaries & Theoretical Analysis

### 2.1 Problem Setup & Kernelization

**LLM Forward Formulation.** Consider an LLM composed of a sequence of $N$ Pre-Norm Transformer layers, $F_1, \ldots, F_N$. Given an input $\mathbf{x}$, for the $l$-th layer, the forward pass is written as

$$\mathbf{h}_{l+1}(\mathbf{x}) = \mathbf{h}_l(\mathbf{x}) + f_l(\text{Norm}(\mathbf{h}_l(\mathbf{x}))). \tag{1}$$

Here, $\text{Norm}(\cdot)$ denotes the Layer Normalization function, $\mathbf{h}_l(\mathbf{x}) \in \mathbb{R}^d$ denotes the $l$-th layer representation and $f_l(\cdot)$ denotes the backbone function of the Transformer layer composed of multi-head attention (MHA) and feed-forward neural network (FFN). For simplicity, we write $\mathbf{h}$ for $\mathbf{h}(\cdot)$ when the context is clear.

**Kernel Model.** A kernel is a function $k(\mathbf{x}, \mathbf{y})$ that measures the similarity between two data points $\mathbf{x}$ and $\mathbf{y}$. It is formally defined as an inner product in a feature space $\mathcal{H}$. Gaussian Radial Basis Function (RBF) kernel (i.e., $k(\mathbf{x}, \mathbf{y}) = \exp(-\gamma \|\mathbf{x} - \mathbf{y}\|^2)$) is the canonical choice due to

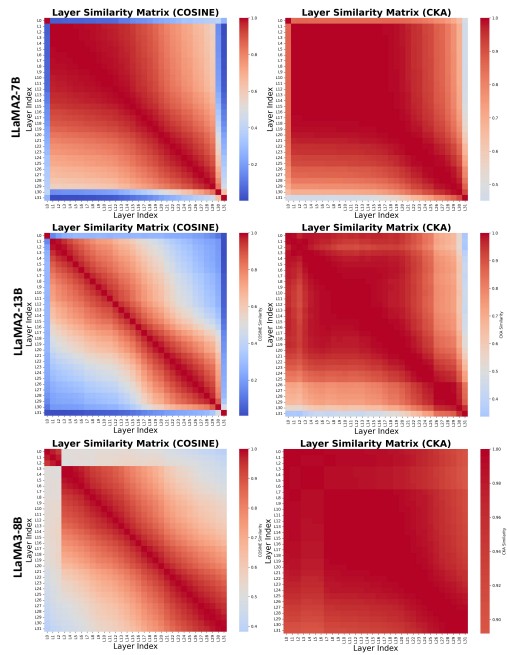

Figure 1: Inter-layer Cosine and CKA Similarity of 3 LLaMA LLMs.

---

**Algorithm 1: Kernelized Dynamics Pruning**

1: **Require:** Pretrained LLM F, calib. set $\mathcal{D}$, RFF dim $m$, low-rank $r$, block max length $K_{\max}$, budget $B$, stability step $\gamma$, weight $W$, training iters $T$.

2: Let $\mathcal{G}$ be the set of $B$ non-overlapping model blocks $F_{l:l+K_{\max}}$ that maximize the similarity score $\mathrm{CKA}(\mathbf{h}_l, \mathbf{h}_{l+K_{\max}})$.

3: **procedure** JOINTKERNELLINEARIZATION
4:    **for** $(l, \cdots, l + K_{\max}) \in \mathcal{G}$ **do**
5:       Define $\varphi_\theta$, operators $\{\mathbf{A}_i\}_{i=1}^k$
6:       **for** minibatch $\mathcal{B} \subset \mathcal{D}$ **do**
7:          $\widehat{\varphi}(\mathbf{h}_{l+k}(\mathcal{B})) \leftarrow \left(\prod_{i=1}^k \mathbf{A}_i\right) \varphi_\theta(\mathbf{h}_l(\mathcal{B}))$
8:          $\mathcal{L}(\cdot) \leftarrow \mathcal{L}_{\mathrm{mse}} + \mathcal{L}_{\cos}$
9:          Update $\theta$ and $\{\mathbf{A}_i\}_{i=1}^k$
10:    **return** $\{\theta^{(l)}, \{\mathbf{A}_i^{(l)}\}_{i=1}^k\}_{l \in \mathcal{G}}$

11: **procedure** TRAININVERSENETWORK
12:    **for** $(l, \cdots, l + K_{\max}) \in \mathcal{G}$ **do**
13:       Define inverse network $\mathcal{R}_\phi$
14:       **for** minibatch $\mathcal{B} \subset \mathcal{D}$ **do**
15:          $\widehat{\mathbf{h}}_{l+k} \leftarrow \mathcal{R}(\widehat{\varphi}(\mathbf{h}_{l+k}))$
16:          $\mathcal{L}(\cdot) \leftarrow \mathcal{L}_{\mathrm{mse}} + \mathcal{L}_{\mathrm{norm}}$
17:          Update $\phi$
18:    **return** $\{\phi^{(l)}\}_{l \in \mathcal{G}}$

19: **procedure** FOLDBLOCK
20:    Define $\mathcal{S}(\mathbf{h}) \leftarrow \mathcal{I}_\phi\left(\left(\prod_{i=1}^k \mathbf{A}_i\right) \varphi_\theta(\mathbf{h})\right)$
21:    Replace $F_l, \ldots, F_{l+k}$ with $\mathcal{S}$

---

its universality and consistency(Steinwart, 2002). In this paper, we employ a learnable Random Fourier Features (RFF) kernel to learn a data-driven, anisotropic Gaussian RBF kernel (Rahimi & Recht, 2007), which enhances representational capacity and reduces computational cost. The kernel is approximated by the inner product of a low-dimensional feature map $\varphi(\cdot) : \mathbb{R}^d \to \mathbb{R}^{2m}$, $\mathrm{k}(\mathbf{x}, \mathbf{y}) \approx \varphi(\mathbf{x})^\top \varphi(\mathbf{y})$, where $\varphi(\mathbf{x})$ is defined as:

$$\varphi(\mathbf{x}) = \frac{1}{\sqrt{m}} \left( \cos\left(\mathbf{W}^\top \mathbf{x} + \mathbf{b}\right)^\top, \sin\left(\mathbf{W}^\top \mathbf{x} + \mathbf{b}\right)^\top \right)^\top. \tag{2}$$

Here the columns of weight matrix $\mathbf{W} \in \mathbb{R}^{d \times m}$ are i.i.d. samples drawn from $\mathcal{N}(0, \boldsymbol{\Sigma})$, and the elements of bias vector $\mathbf{b} \in \mathbb{R}^m$ are i.i.d. samples drawn from $\mathrm{U}(0, 2\pi)$. In contrast to the original RFF kernel, which relies on a pre-defined spectral distribution, here the covariance matrix $\boldsymbol{\Sigma}$ is learnable and parameterized as $\boldsymbol{\Sigma} = \mathbf{D} + \mathbf{L}\mathbf{L}^\top$, $\mathbf{D} = \mathrm{diag}(\exp(\boldsymbol{\lambda}))$ is a diagonal matrix parameterized by a learnable vector $\boldsymbol{\lambda} \in \mathbb{R}^d$ and $\mathbf{L} \in \mathbb{R}^{d \times r}$ is a learnable low-rank factor matrix, where the rank $r$ is a hyperparameter that satisfies $r \ll d$. Equation 2 effectively learns an anisotropic Guassian RBF kernel $\mathrm{k}(\mathbf{x}, \mathbf{y}) = \exp\left(-\frac{1}{2}(\mathbf{x} - \mathbf{y})^\top \boldsymbol{\Sigma}^{-1}(\mathbf{x} - \mathbf{y})\right)$ whose distance metric is adapted to the data.

**Symbol Denotation.** In this paper, the $\ell_p$-norm of a vector $\mathbf{v} = (v_1, \cdots, v_m)^\top$ is denoted by $\|\mathbf{v}\|_p \triangleq \sum_{i=1}^m (\mathbf{v}_i^p)^{1/p}$, with the subscript omitted for the standard $\ell_2$-norm (i.e., $\|\mathbf{v}\| \triangleq \|\mathbf{v}\|_2$), the operator norm for a matrix is denoted by $\|\cdot\|_{\mathrm{op}}$, $(x)_+ = \max\{x, 0\}$ denotes the positive part of vairable $x$, and $O(\cdot)$ denotes the standard Big-O notation for an asymptotic upper bound. For detailed definitions of technical machine learning terms, such as ERM and population risk, please refer to the Appendix A.

## 2.2 FORWARD DYNAMICS ON A SLOW MANIFOLD

The residual connection described in Equation 1 motivates us to view the Transformer's forward pass as a discrete-time dynamical system. The term $\mathrm{f}_l(\cdot)$ can be seen as a "velocity" or "update" vector that perturbs the current state $\mathbf{h}_l$, thereby producing the state of the next layer, $\mathbf{h}_{l+1}$.

As shown in Figure 1, consecutive layer representations in LLMs exhibit strikingly high similarity. Within this dynamical framework, high similarity between $\mathbf{h}_{l+1}$ and $\mathbf{h}_l$ is directly equivalent to the update vector having a small relative norm, i.e., $\|\mathrm{f}_l(\mathrm{Norm}(\mathbf{h}_l))\| \ll \|\mathbf{h}_l\|$, indicating that the

system's trajectory has entered a "slow manifold". If the transformation between layers is minor, the local dynamics of a Transformer block $f_l$ could potentially be approximated by a far simpler function. This is a common practice in fields such as the numerical analysis of partial differential equations (PDEs) and model order reduction (e.g. (Murray, 2007), more detailed analysis can be found in the Appendix H). While a first-order approximation (i.e., local linearization of the dynamics) is a natural approach, the Transformer's forward pass is fundamentally nonlinear. Applying such an approximation directly in the original representation space would therefore lead to significant information loss, as it would neglect crucial nonlinear feature interactions. Note that in Figure 1, the CKA similarity generally exceeds cosine similarity, which is consistent with the property that the high-dimensional representation induced by the kernel space captures complex representational relationships more effectively. Therefore, we assume that there exists a kernel function $\varphi(\cdot)$ that induces a Hilbert space where the layer-wise transformation becomes approximately linear, i.e., $\varphi(\mathbf{h}_{l+1}) \approx \mathbf{A}_l \varphi(\mathbf{h}_l)$.

## 2.3 THEORETICAL RESULTS

As argued in Section 2.2, when adjacent layers exhibit high representational similarity, the forward propagation enters a slow manifold where a simpler rule can capture short-horizon dynamics, an approximation that becomes more accurate in a suitable kernel space. In this section, we present the theoretical formalization of these arguments. First, we derive the error bound for linear fitting of multi-layer representations in the kernel space, demonstrating that consecutive layers with highly similar representations can be reduced to linear transformations in this space. Subsequently, we prove that the kernel space exhibits superior fitting capacity for layer simplification compared to the original representation space. The detailed proofs are provided in Appendix D.

**Theorem 1** (Kernel Linearization Error Bound). *Let $\varphi_\theta : \mathbb{R}^d \to \mathbb{R}^{2m}$ be a feature map such that $\|\varphi_\theta(\mathbf{h})\|_2 \leq R_\varphi$ for all representations $\mathbf{h}$. Let $\mathbf{A}_i$ be linear operators for the $i$-th step transition with $\|\mathbf{A}_i\|_{\mathrm{op}} \leq B_\mathbf{A}$. Let $(\widehat{\theta}, \{\widehat{\mathbf{A}}_i\})$ be the Empirical Risk Minimization(ERM) solution on an $n$-sample training set, and let $L_{\mathrm{ERM}}$ denote the minimum empirical one-step squared error. Then for any $\delta \in (0, 1)$, with probability at least $1 - \delta$ over the draw of the training set, for every starting layer $l$ and horizon $k \geq 1$, the $k$-step error $\mathcal{E}_{k,l}(\widehat{\theta}, \{\widehat{\mathbf{A}}_i\}) := \varphi_{\widehat{\theta}}(\mathbf{h}_{l+k}) - \left( \prod_{i=l}^{l+k-1} \widehat{\mathbf{A}}_i \right) \varphi_{\widehat{\theta}}(\mathbf{h}_l)$ satisfies*

$$\left\| \mathcal{E}_{k,l}(\widehat{\theta}, \{\widehat{\mathbf{A}}_i\}) \right\| \leq \underbrace{\sqrt{L_{\mathrm{ERM}} + C\, B_\mathbf{A}^2 R_\varphi^2 \sqrt{\frac{2m \log(2m/\delta)}{n}}}}_{(a)} \cdot \underbrace{\sum_{j=0}^{k-1} B_\mathbf{A}^{k-1-j}}_{(b)} . \qquad (3)$$

Theorem 1 yields a two-factor bound on $k$-step error: a single-step term $(a)$ that captures the linear model's in-kernel fitting ability plus its generalization gap, and a multi-step accumulation factor $(b)$ that governs how errors accumulate across replaced layers. Theoretically, for the bounded parameters $R_\varphi$ and $B_\mathbf{A}$, the error bound converges at a rate of $O(1/\sqrt{n})$ with the sample size $n$. In subsequent experiments illustrated in Figure 5, we establish empirical values for $B_\mathbf{A}$ and $R_\varphi$, demonstrating that these bounds are well-behaved, i.e., they are significantly smaller than $\sqrt{n}$. Therefore, Theorem 1 demonstrates that consecutive layers can be well simplified as linear transformations in kernel space.

Further, we can prove that compared to learning a linear approximation in the original representation space, the kernel space yields a smaller linear approximation error. Before that, the fitting capability of kernel space is given in the following lemma.

**Lemma 1** (Kernel Existence). *For any given $\epsilon > 0$, there exists a Random Fourier Feature map $\varphi : \mathbb{R}^d \to \mathbb{R}^{2m}$(defined by suitable parameters $\mathbf{W}, \mathbf{b}$, and a sufficiently large dimension $2m$) and a corresponding linear operator $\mathbf{A}$, such that: $\mathbb{E}_{\mathbf{h}_l \sim \mathcal{D}} \left[ \|\varphi(\mathbf{h}_{l+1}) - \mathbf{A}\varphi(\mathbf{h}_l)\| \right] < \epsilon$.*

The proof of Lemma 1 is a direct application of the Universal Kernels (Micchelli et al., 2006). Next, our analysis begins with two function classes. Consider the following two function classes at the $l$-th layer of an LLM: the identity class of the original space $\mathcal{F}_{\mathrm{ID}} := \{x \mapsto \mathbf{A}x : \|\mathbf{A}\|_{\mathrm{op}} \leq B_\mathbf{A}\}$, with a population risk of $\mathcal{R}_{\mathrm{ID}}(\mathbf{A}) := \mathbb{E}\left[\|\mathbf{h}_{l+1} - \mathbf{A}\mathbf{h}_l\|^2\right]$; the RFF class $\mathcal{F}_{\mathrm{RFF}} := \{x \mapsto \mathbf{A}\varphi_\theta(x) : \|\mathbf{A}\|_{\mathrm{op}} \leq B_\mathbf{A}, \|\varphi(x)\| \leq R_\varphi, \theta \in \Theta\}$, with a population risk of

$\mathcal{R}_{\mathrm{RFF}}(\theta, \mathbf{A}) := \mathbb{E}\left[\|\varphi_\theta(\mathbf{h}_{l+1}) - \mathbf{A}\varphi_\theta(\mathbf{h}_l)\|^2\right]$. Note the minimum population approximation error of the two classes as:

$$\mathrm{apx}_{\mathrm{ID}} := \inf_{\|\mathbf{A}\|_{\mathrm{op}} \leq B_\mathbf{A}} \mathcal{R}_{\mathrm{ID}}(\mathbf{A}), \qquad \mathrm{apx}_{\mathrm{RFF}} := \inf_{\theta \in \Theta, \|\mathbf{A}\|_{\mathrm{op}} \leq B_\mathbf{A}, \|\varphi(x)\| \leq R_\varphi} \mathcal{R}_{\mathrm{RFF}}(\theta, \mathbf{A}).$$

Let $\widehat{\mathbf{A}}_{\mathrm{ID}}$ and $(\widehat{\theta}, \widehat{\mathbf{A}}_{\mathrm{RFF}})$ be the ERM solutions on the respective classes. We can get the following conclusion on the superior fitting capacity of Kernel space.

**Theorem 2** (Kernel Advantage). *Let* $\Delta := \mathrm{apx}_{ID} - \epsilon > 0$. *For any* $\delta$, *there exist* $m', c, C > 0$, *such that for RFF dimension* $2m > m'$ *and*

$$n \geq \frac{16\left(CB_\mathbf{A}^2\left(R_\varphi^2\sqrt{2m\log\frac{2cm}{\delta}} - R_h^2\sqrt{d\log\frac{cd}{\delta}}\right)_+\right)^2}{\Delta^2}, \tag{4}$$

*the following holds with probability at least* $1 - \delta$: $\mathcal{R}_{RFF}(\widehat{\theta}, \widehat{\mathbf{A}}_{RFF}) \leq \mathcal{R}_{ID}(\widehat{\mathbf{A}}_{ID})$.

Theorem 2 establishes that in the kernel space, a well-trained RFF with dimensionality greater than $2m$ achieves lower population risk compared to simplification in the original representation space. This result further substantiates the necessity of performing linearized model compression in the kernel space. Experiments in Table 3 empirically validate the superiority of the kernel space over the original representation space.

## 3 KERNELIZED DYNAMICS PRUNING

As argued in Section 2, on both intuitive and theoretical grounds, representational similarity implies computational redundancy, so that the complex module can be simplified better in kernel space. Building on this foundation, this section details our pruning method, Kernelized Dynamics Pruning (KDP). The algorithm is presented in Algorithm 1.

### 3.1 KERNEL LINEARIZATION JOINT TRAINING

The first step is to identify which parts of the network to prune. Inspired by prior work (Chen et al., 2025; Ding et al., 2025; Gromov et al., 2025), we focus on replacing blocks of multiple consecutive Transformer layers. To control the cumulative error as shown in Equation 1, we set the consecutive layers to be replaced with a maximum length $K_{\max}$. For each block, we first compute the CKA similarity between the output representations of the first and last layers of consecutive blocks. Following standard practice (e.g., Ding et al., 2025), we exclude the initial and final 10% of the model's layers from consideration, as they are empirically known to be sensitive to pruning. Combining these criteria, we rank all eligible blocks according to their CKA scores and advance the highest-scoring candidate to the kernel linearization training stage.

For the candidate block for pruning, we learn a replacement module by jointly optimizing the learnable RFF kernel parameters $\theta$ and the multi-step linear operators $\{\mathbf{A}_i\}_{i=1}^{K_{\max}}$. For each training sample $x \in \mathcal{D}$, the estimate $(\widehat{\theta}, \{\widehat{\mathbf{A}}_i\})$ is given by:

$$\arg\min_{\theta, \{\mathbf{A}_i\}} \sum_{i=1}^{K_{\max}} \sum_{x \in \mathcal{D}} \left[\|\mathbf{A}_i\varphi_\theta\left(\mathbf{h}_{l+i-1}(x)\right) - \varphi_\theta\left(\mathbf{h}_{l+i}(x)\right)\|^2 \right. \tag{5}$$
$$\left. + \left(1 - W \odot \cos\left(\mathbf{A}_i\varphi_\theta\left(\mathbf{h}_{l+i-1}(x)\right), \varphi_\theta\left(\mathbf{h}_{l+i}(x)\right)\right)\right)\right].$$

Here, $\odot$ denotes element-wise multiplication. The loss function is composed of two terms: a reconstruction loss and a weighted cosine similarity loss. The latter is designed to encourage a finer-grained alignment of the representations' geometric structure. Specifically, this cosine similarity term is modulated by a weight matrix $W$, which assigns greater importance to tokens based on their position, prioritizing those towards the end of the sequence. To improve convergence stability, $\mathbf{A}_i$ is parameterized as $\mathbf{A}_i = \mathbf{I} + \gamma_i\mathbf{B}_i$ that preserves the additive nature of LLM's forward pass (see Equation 1), where the scalar $\gamma_i$ and the matrix $\mathbf{B}_i$ are trainable parameters. In addition, we initialize $\mathbf{A}_i$ at the start of each iteration with its Ordinary Least Squares (OLS) estimate in the current RFF kernel space to accelerate training.

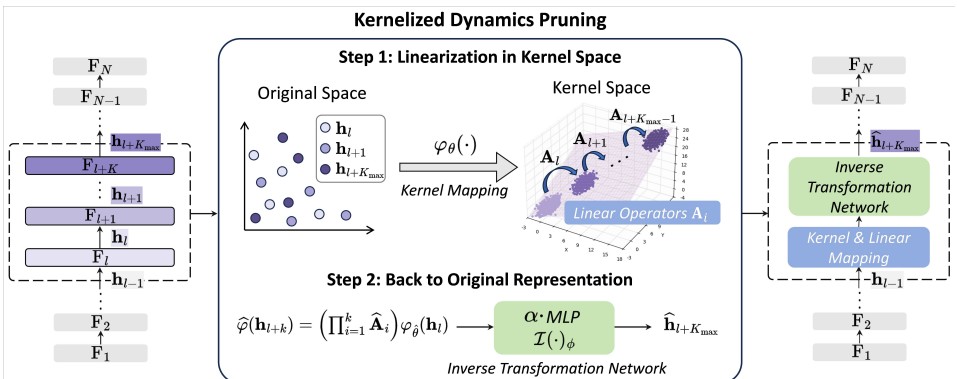

Figure 2: Overall illustration of KDP. KDP replaces a multi-layer Transformer block via a two-step process: first, it projects and linearizes non-linear representations in a kernel space; second, it learns an inverse mapping from the kernel space back to the original space using a simple network.

## 3.2 INVERSE TRANSFORMATION NETWORK

Once the joint training converges, we can get $k$-step prediction in the kernel space, i.e., $\widehat{\varphi}(\mathbf{h}_{l+k}) = \left(\prod_{i=1}^{k} \widehat{\mathbf{A}}_i\right) \widehat{\varphi}_\theta(\mathbf{h}_l)$. However, this output must be mapped back to the original space for layer replacement. Thus, an inverse transformation network, $\mathcal{I}(\cdot)_\phi : \mathbb{R}^{2m} \to \mathbb{R}^d$, takes the kernel-space prediction $\widehat{\varphi}(\mathbf{h}_{l+k})$ as input, to reconstruct representation in original space $\mathbf{h}_{l+k}$, denote as $\widehat{\mathbf{h}}_{l+k}$. The estimate $\widehat{\phi}$ is given by:

$$\arg\min_\phi \sum_{x \in \mathcal{D}} \|\mathcal{I}_\phi\left(\widehat{\varphi}\left(\mathbf{h}_{l+k}\left(x\right)\right)\right) - \mathbf{h}_{l+k}(x)\|^2. \tag{6}$$

We parameterize the inverse transformation network $\mathcal{I}$ as a two-layer MLP with a scalar scaling factor $\alpha$, i.e. $\mathcal{I}(\mathbf{x}) := \alpha \cdot \mathrm{MLP}(\mathbf{x})$. We observe that the iterative application of the linear operators $\{\mathbf{A}_i\}$ in the kernel space can lead to a significant decrease in the representation's norm. The scaling factor $\alpha$ explicitly compensates for this norm decay, enabling a more stable and efficient recovery of the original representation's scale.

## 4 EXPERIMENTS

### 4.1 SETUP

**Models.** Our experiments are conducted on 6 popular open-source models: LLaMA2-7B, LLaMA2-13B (Touvron et al., 2023), LLaMA3.1-8B, LLaMA3-8B (Grattafiori et al., 2024), OPT-2.7B and OPT-6.7B. (The results for the OPT model are presented in the Appendix E.) Following prior work, we prune the layers that account for approximately 25% of the original parameters.

**Training Datasets.** For training, we construct a composite dataset of 4000 samples by randomly selecting 500 samples from the training set of each of the following datasets: PIQA (Bisk et al., 2020), CMMLU (Yüksel et al., 2024), BoolQ (Clark et al., 2019), C3 (Sun et al., 2020), MNLI (Williams et al., 2018), Race-High/Middle (Lai et al., 2017), and SlimPajama (Shen et al., 2024).

**Benchmarks.** For a comprehensive evaluation, we assess the models on 15 standard datasets. We group the evaluation tasks into two categories: generation and classification. For the classification, the evaluation is conducted on the test sets of the aforementioned datasets, excluding SlimPajama, and is supplemented with 6 additional datasets: CHID (Zheng et al., 2019), HellaSwag (Zellers et al., 2019), CoQA (Reddy et al., 2019), WSC (Levesque et al., 2012), SST2 (Socher et al., 2013), and C3 (Sun et al., 2020). We also evaluate 3 benchmarks for generative capabilities. For WikiText (Merity et al., 2017) and C4 (Raffel et al., 2020), we evaluate their Perplexity (PPL), and for XSum (Narayan et al., 2018), we evaluate ROUGE1. We use the LM Evaluation Harness (Gao et al., 2024) with default parameters in our evaluations.

**Baselines.** Beyond the dense model, we consider six structured pruning methods: ShortGPT (Men et al., 2024) and SLEB (Song et al., 2024), which prune the network directly via metrics without

retraining, and LaCo (Yang et al., 2024), Streamline (Chen et al., 2025), SliceGPT (Ashkboos et al., 2024), and LLM-Pruner (Ma et al., 2023), which first prune or replace network components and then employ a retraining phase to recover performance.

## 4.2 MAIN RESULTS

Table 1 and Table 2 show the effects of different pruning methods on classification and generative tasks. The two tables present detailed scores on each benchmark, the overall average, and the retained performance rate. In the tables, "Dense" refers to the original model and "w/o kernel" is an ablation of our method that only removes the identified layers. "Ours" denotes our proposed method, while "Ours†" refers to our method after being fine-tuned using Parameter-Efficient Fine-Tuning (PEFT). (Details of PEFT are provided in the Appendix B.) Table 1 shows that our method's retained performance surpasses that of the best-performing baseline by 9.1%, 8.3%, and 9.3% on LLaMA2-7B, LLaMA2-13B, and LLaMA3-8B, respectively. Our method also achieves comparable performance on generative tasks. Notably, it requires no retraining phase to recover performance.

It is worth noting that on the SST-2 dataset, a simple binary classification task, existing methods show a notable performance degradation despite performing well on other complex benchmarks. We suppose that this phenomenon arises because simpler tasks rely more heavily on coarse-grained information, which prior methods tend to prune inadvertently, thereby leading to instability. In contrast, our method consistently maintains its effectiveness, indicating its superior capacity to preserve the model's essential capabilities.

The comparison with the "w/o Kernel" demonstrates the effectiveness of KDP. While directly pruning consecutive layers leads to a significant degradation in performance, our method of simplifying these layers in the kernel space yields substantial gains. Specifically, it boosts the average retained performance by 31.9%, 23.1%, and 18.9% percentage points for the three models, respectively.

## 4.3 FURTHER ANALYSIS

**Representational patterns can be learned within the kernel space.** Figure 4 presents a comparison between the predicted representations and the ground-truth representations viewed in the kernel space. Specifically, we feed a sample from the CMMLU dataset into LLaMA2-7B, where layers 10 through 14 have been replaced by KDP. RFF kernel projects the $14\text{-}th$ layers' output representations into a (sequence length $\times$ 1024) matrix. To visualize these representations, we take rows 0, 25, and 50, reshape each into a $32 \times 32$ matrix, and render them as heatmaps. As observed, minor deviations tend to increase with the sequence position, which motivated our weight-scaling design in Equation 5. To this end, the predicted representations closely match the actual ones. Notably, KDP not only captures the general trends of the representations but also accurately fits outlier points, which are critical to model performance (Sun et al., 2024). This highlights KDP's ability to effectively learn and preserve essential patterns of the representations in the kernel space.

**As training progresses, $B_{\mathbf{A}}$ and $R_{\varphi}$ behave well.** Figure 5 illustrates the evolution of the operator norms for $\{\mathbf{A}_i\}$ and $\varphi(\cdot)$ during the training of two LLMs. $R_{\varphi}$ remains consistently below 1.5, while $B_{\mathbf{A}}$ decreases rapidly before converging. These observations validate the effectiveness of the error bound presented in Theorem 1.

**Step 1 dominates the pruning performance, while Step 2 serves primarily as the inverse kernel mapping.** Figure 4 intuitively illustrates the effectiveness of kernel-space fitting in Step 1. To gain deeper insights into the KDP process and identify the component responsible for high performance, we analyze its training dynamics in Figure 3. The left panel shows that in Step 1, the loss drops sharply and converges within approximately 100 epochs. Concurrently, the middle panel depicts the cosine similarity between the predicted and ground-truth representations increasing as the loss decreases, confirming effective learning in the kernel space. By contrast, the right panel shows that

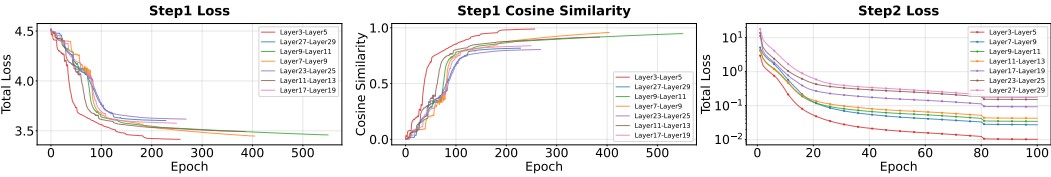

Figure 3: Training loss on LLaMA2-13B. From left to right, the panels show: (a) the loss curve during Step 1, (b) the cosine similarity during Step 1, and (c) the loss curve for Step 2.

Table 1: Performance comparison of different baselines on classification benchmarks. "*" indicates the results in the original paper. "†" indicates the results after fine-tuning. Average (Avg.) represents the arithmetic mean of Accuracy across all datasets. Retained performance (RP.) represents the percentage of the original model's performance retained by the pruning method.

| LLM | Method | Ratio | CMNLI | HeSw | PIQA | CHID | WSC | CoQA | BoolQ | MMLU | CMMLU | Race | SST2 | C3 | Avg. | RP. |
|---|---|---|---|---|---|---|---|---|---|---|---|---|---|---|---|---|
| LLaMA-2-7B | Dense | 0.0% | 34.9 | 73.3 | 79.7 | 41.6 | 81.2 | 66.2 | 72.3 | 48.9 | 30.8 | 42.8 | 93.2 | 44.8 | 59.14 | 100.0 |
| | SLEB | 20.1% | 29.7 | 63.3 | 66.1 | 19.3 | **75.7** | 48.5 | 68.2 | 24.2 | 26.4 | 28.8 | 78.8 | 35.3 | 47.02 | 79.5 |
| | ShortGPT | 27.1% | 31.3 | 51.6 | 66.3 | 25.5 | 70.9 | **49.0** | 59.6 | 34.4 | 28.5 | 37.7 | 68.4 | 35.9 | 46.59 | 78.8 |
| | LaCo* | 27.1% | 34.4 | 55.7 | 69.8 | **36.1** | 40.4 | 45.7 | 64.1 | 26.5 | 25.2 | 23.6 | - | 39.7 | 41.93 | 70.9 |
| | Streamline† | 27.0% | 32.9 | 54.3 | 63.6 | 26.9 | 70.7 | 42.5 | 64.3 | 37.7 | **29.5** | 34.9 | 90.5 | 32.6 | 48.37 | 81.8 |
| | LLMPruner† | 24.8% | 31.4 | 55.4 | **70.1** | 27.8 | 71.4 | 44.7 | 53.3 | 25.7 | 25.1 | 25.4 | 67.7 | 23.1 | 43.43 | 73.4 |
| | SliceGPT | 25.4% | 31.9 | 49.9 | 68.7 | 16.7 | 66.5 | 48.6 | 55.5 | 28.2 | 22.3 | 25.2 | 78.3 | 30.7 | 43.54 | 73.6 |
| | SliceGPT† | 25.4% | 32.1 | 53.3 | 69.2 | 20.4 | 61.2 | 44.4 | 61.9 | 29.7 | 22.5 | 26.7 | 87.1 | 28.7 | 44.77 | 75.7 |
| | w/o Kernel | 24.8% | 30.8 | 34.5 | 57.1 | 12.8 | 69.1 | 11.2 | 50.3 | 25.3 | 24.8 | 20.5 | 51.5 | 23.5 | 34.28 | 58.0 |
| | Ours | 22.8% | 33.6 | **65.1** | 70.1 | 27.2 | 73.9 | 45.4 | 71.6 | **44.7** | 28.1 | 39.9 | **94.0** | **43.7** | **53.11** | **89.9** |
| | Ours† | 22.8% | **34.6** | 63.0 | 68.8 | 29.5 | 74.8 | 49.0 | 72.3 | 41.5 | 26.7 | 40.4 | 92.3 | 37.3 | 52.52 | 88.8 |
| LLaMA-2-13B | Dense | 0.0% | 48.1 | 74.4 | 77.5 | 47.9 | 88.5 | 65.6 | 73.9 | 57.7 | 38.9 | 59.7 | 93.7 | 47.5 | 64.45 | 100.0 |
| | SLEB | 19.5% | 34.7 | 63.1 | 67.2 | 39.0 | 77.7 | 51.7 | 52.1 | 25.6 | 24.6 | 53.5 | 77.9 | 31.7 | 49.9 | 77.4 |
| | ShortGPT | 24.6% | 31.7 | 61.5 | 72.5 | 38.4 | 70.0 | 51.0 | 62.8 | 54.4 | 33.7 | 57.3 | 76.1 | 45.0 | 54.53 | 84.6 |
| | LaCo* | 24.6% | 32.9 | 64.4 | 74.3 | 40.1 | 52.9 | 52.7 | 64.0 | 45.9 | 32.6 | 55.6 | - | 44.9 | 50.94 | 79.0 |
| | Streamline* | 24.6% | 33.0 | **69.1** | **75.1** | 38.0 | 36.5 | **63.8** | 66.2 | 55.1 | **39.2** | 58.0 | - | 45.7 | 52.70 | 81.8 |
| | LLMPruner† | 24.4% | 27.8 | 55.3 | 61.9 | 30.7 | 63.4 | 49.7 | 53.0 | 23.1 | 23.3 | 20.0 | 65.8 | 30.3 | 42.03 | 65.2 |
| | SliceGPT | 23.6% | 33.2 | 53.1 | 60.8 | 18.4 | 74.8 | 44.3 | 36.6 | 25.1 | 25.7 | 22.4 | 68.3 | 26.1 | 40.73 | 63.5 |
| | SliceGPT† | 23.6% | 31.5 | 45.9 | 60.3 | 18.9 | 67.1 | 39.7 | 37.8 | 29.1 | 28.4 | 23.4 | 78.0 | 26.9 | 40.58 | 63.0 |
| | w/o Kernel | 24.3% | 32.8 | 63.1 | 67.5 | 33.6 | 72.5 | 19.3 | 60.6 | 35.1 | 27.1 | 33.3 | 61.7 | 33.5 | 45.01 | 69.8 |
| | Ours | 23.4% | 40.0 | 64.1 | 74.5 | 39.1 | 82.5 | 62.5 | 69.4 | 52.7 | 37.1 | 58.2 | 89.1 | 49.5 | 59.89 | 92.9 |
| | Ours† | 22.8% | **41.2** | 63.5 | 73.9 | 40.2 | 80.1 | 63.8 | 63.3 | 51.9 | 38.0 | 55.6 | 87.8 | 48.5 | 58.98 | 91.5 |
| LLaMA-3-8B | Dense | 0.0% | 33.9 | 75.7 | 77.9 | 72.8 | 86.4 | 72.5 | 75.3 | 67.2 | 50.1 | 73.5 | 93.0 | 62.1 | 70.03 | 100.0 |
| | SLEB | 19.0% | 33.0 | **65.1** | 63.0 | **28.0** | 74.6 | 31.5 | 67.7 | 49.3 | 27.3 | 24.1 | 77.9 | 37.2 | 48.23 | 68.9 |
| | ShortGPT | 19.0% | 33.7 | 46.0 | 65.4 | 25.8 | 73.0 | **44.8** | 38.9 | 34.2 | 37.1 | 30.4 | 82.3 | 44.1 | 46.31 | 66.1 |
| | w/o Kernel | 27.2% | 33.0 | 41.0 | 55.7 | 18.0 | 73.8 | 29.3 | 50.8 | 24.0 | 25.7 | 21.7 | 65.1 | 36.8 | 39.58 | 56.5 |
| | Ours | 25.8% | **33.9** | 61.5 | 73.5 | 23.9 | 83.6 | 37.7 | 71.0 | 55.5 | 34.0 | 38.4 | 81.4 | 38.9 | 52.77 | 75.4 |
| | Ours† | 22.8% | 30.1 | 57.5 | **74.1** | 25.1 | **84.4** | 33.0 | 62.4 | 50.4 | 38.5 | 34.0 | 68.3 | 35.5 | 49.44 | 70.6 |

Table 2: Performance comparison of different baselines on generation benchmarks. For Perplexity, a lower score means better performance (↓), while for ROUGE, a higher score means better (↑).

| Models | LLaMA2-7B | | | LLaMA2-13B | | | LLaMA-3.1-8B | | | RP(PPL) | RP(Rouge) |
|---|---|---|---|---|---|---|---|---|---|---|---|
| | WIKI↓ | C4↓ | Xsum↑ | WIKI↓ | C4↓ | Xsum↑ | WIKI↓ | C4↓ | Xsum↑ | | |
| Dense | 5.5 | 7.0 | 19.2 | 4.9 | 6.5 | 22.7 | 6.1 | 8.8 | 27.2 | 100% | 100% |
| Streamline† | 9.9 | 17.1 | **19.7** | 43.7 | 81.7 | **19.5** | 377.1 | 201.1 | 21.5 | 5.3% | **87.8%** |
| SLEB | 12.9 | 19.3 | 10.0 | 33.3 | 45.5 | 17.3 | 55.8 | 79.6 | 13.1 | 15.7% | 58.5% |
| w\o Kernel | 13.2 | 17.4 | 12.8 | 20.7 | 27.9 | 16.0 | 120.7 | 103.6 | 17.5 | 12.8% | 67.0% |
| Ours | **8.3** | 13.5 | 15.3 | 17.2 | 25.0 | 18.0 | 205.7 | 113.6 | **21.8** | 10.1% | 79.7% |
| Ours† | **8.3** | **9.3** | 17.0 | **13.2** | **15.0** | 19.3 | 31.7 | 14.6 | 20.2 | **42.1%** | 81.8% |

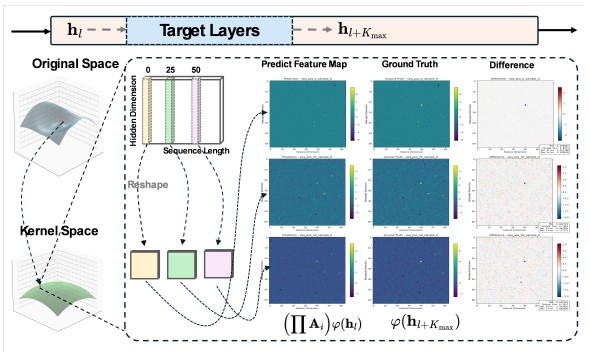

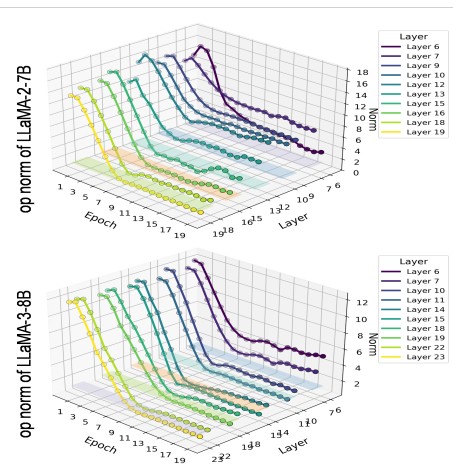

Figure 4: Kernel space representation visualization. The heatmap panels show (from left to right) the predicted representation, the ground-truth representation, and their difference for a sample from CMMLU. The color range is unified for the first two panels to highlight their similarity.

Figure 5: Operator norms of $\varphi(\cdot)$ and $\{\mathbf{A}_i\}$ during training. The line corresponds to $\|\mathbf{A}_i\|_{\text{op}}$; the shaded region is the norm of $\|\varphi(\bar{\mathbf{h}}_l)\|_{\text{op}}$ for the shared kernel space, calculated from mean-subtracted samples.

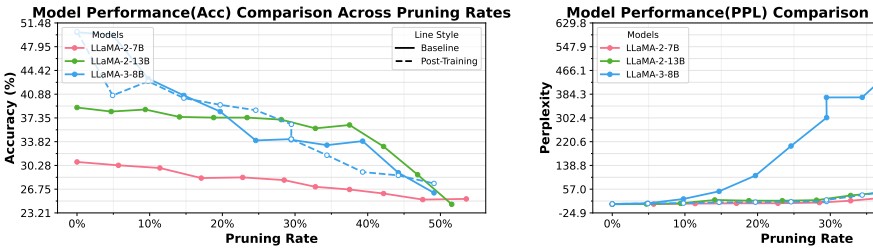

Figure 6: Performance of CMMLU and WIKI when we prune with increasing pruning ratio.

the loss in Step 2 converges within only 20 steps, suggesting that the inverse transformation network $\mathcal{I}$ carries out a relatively simple task. We also conduct an ablation study to evaluate the performance impact of barely training for each step, with the results detailed in the Appendix F. In conclusion, these results demonstrate that Step 1 is the primary contributor to the pruning performance of KDP.

## 4.4 ABLATION STUDY

**Different Pruning Ratios.** As shown in Figure 6, we evaluate our method on the CMMLU and WIKI Text across various models and pruning ratios. The method exhibits strong retained performance on CMMLU, maintaining over 90% of the original performance at a pruning rate of around 28% and over 80% at rates exceeding 50%, which confirms its robustness. The results show that the models' overall performance degrades linearly with an increasing pruning rate. However, on generative tasks, the PPL behaves differently across models. Specifically, LLaMA3-8B exhibits a rapid increase, whereas the PPL for LLaMA2-7B and LLaMA2-13B remains relatively stable.

**Different Linearization Methods.** To validate the importance of the kernel transformation, we compare KDP against a simplified baseline that performs linear fitting directly in the original representation space. As shown in Table 3, both OLS-based methods exhibit a drastic degradation in performance on the LLaMA3-8B model. In contrast, KDP significantly outperforms both baselines across all tasks. These results provide strong evidence that the transformations between Transformer layers are highly non-linear. Linear approximation in the original representation space, as attempted by OLS, is incapable of capturing these complex dynamics, resulting in severe information loss. This observation further corroborates Theorem 2, demonstrating the necessity of learning linear transformations in the kernel space.

**Different Max Length.** To analyze the impact of $K_{\max}$, we incrementally prune more layers from LLaMA2-7B and compare two strategies for block selection: expanding a fixed initial block or choosing a new one based on CKA similarity. Table 4 demonstrates our method's robustness in simplifying short-term dynamics. Model performance is stable when replacing few layers ($K_{\max} \leq 3$) but degrades catastrophically as more layers are replaced. This observed non-linear degradation strongly supports our theoretical analysis (Theorem 1), which shows that the approximation error grows exponentially with the number of layers, leading to instability.

Table 3: Comparison with direct linearization in the original space via OLS on LLaMA3-8B.

| Method | HeSW | CoQA | MMLU |
|---|---|---|---|
| Dense | 75.7 | 72.5 | 67.2 |
| Direct OLS | 31.3 | 0.0 | 26.7 |
| Sequential OLS | 30.7 | 0.0 | 28.2 |
| Ours | 61.5 | 37.7 | 55.5 |

Table 4: Ablation study on max length $K_{\max}$.

| $K_{\max}$ | Pruned Layers | HeSW | CoQA | MMLU |
|---|---|---|---|---|
| 3 | [6,7] | 72.1 | 65.2 | 48.5 |
| 4 | [6,7,8] | 71.0 | 64.8 | 46.0 |
| 4 | [10,11,12] | 71.3 | 65.1 | 47.7 |
| 5 | [6,7,8,9] | 65.1 | 57.8 | 43.8 |
| 5 | [12,13,14,15] | 66.1 | 61.6 | 44.9 |
| 6 | [6,7,8,9,10] | 47.6 | 41.8 | 24.0 |
| 6 | [11,12,13,14,15] | 43.2 | 40.4 | 27.3 |

## 5 CONCLUSION

In this paper, we introduce a new perspective on layer pruning that simplifies the information flow by operating within a kernel space. Based on this perspective, we propose a novel method named as KDP. We demonstrate the effectiveness of our method through both theoretical analysis and empirical experiments. Our future work includes exploring more kernel functions and adapting our pruning method for multimodal LMs.

## REPRODUCIBILITY STATEMENT

We have taken several measures to ensure the reproducibility of our work. A detailed description of our proposed framework, algorithms, and experimental settings is provided in the main text and Appendix B. Additional implementation details and ablation studies are included in the paper. For theoretical results, we present all assumptions and complete proofs in the Appendix. All datasets used in our experiments are publicly available. Furthermore, we submit an anonymous link to the source code and scripts for reproducing our experiments.

## ACKNOWLEDGMENTS

Wu gratefully acknowledges the financial support from the National Natural Science Foundation of China (Grant No. 72173086).

This work is also funded by Science and Technology Commission of Shanghai Municipality Program, China (No.24DZ2202100) and Shanghai Municipal Commission of Economy and Informatization, China (No.RZ-RGZN-01-25-1369).

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

APPENDIX

## A    PRELIMINARIES

**Empirical Risk Minimization.** Empirical Risk Minimization (ERM) is a fundamental principle in statistical learning theory. Given a class of functions $\mathcal{H}$ and a dataset $\mathcal{D} = \{(x_i, y_i)\}_{i=1}^n$ sampled i.i.d. from an unknown distribution $\mathcal{P}$, ERM selects the function that minimizes the empirical loss:

$$\hat{R}(h) = \frac{1}{n} \sum_{i=1}^n \ell(h(x_i), y_i), \qquad h_{\text{ERM}} = \arg\min_{h \in \mathcal{H}} \hat{R}(h).$$

ERM serves as a tractable surrogate for minimizing the population risk $R(h) = \mathbb{E}_{(x,y) \sim \mathcal{P}} [\ell(h(x), y)]$.

**Population Approximation Error.** The Population Approximation Error (PAE) measures the discrepancy between empirical performance and the true expected performance:

$$\text{apx}(h) = \big| R(h) - \hat{R}(h) \big|.$$

A small approximation error indicates that the empirical distribution is a good proxy for the underlying population distribution.

**$L'$-Lipschitz Continuity.** A function $f : \mathcal{X} \to \mathbb{R}^d$ is $L'$-Lipschitz continuous if

$$\|f(x) - f(x')\| \le L' \|x - x'\|, \qquad \forall x, x' \in \mathcal{X}.$$

**Rademacher Complexity.** Given a class of real-valued functions $\mathcal{F}$ and samples $\{x_1, \ldots, x_n\}$, the empirical Rademacher complexity of $\mathcal{F}$ is defined as

$$\hat{\mathfrak{R}}_n(\mathcal{F}) = \mathbb{E}_\sigma \left[ \sup_{f \in \mathcal{F}} \frac{1}{n} \sum_{i=1}^n \sigma_i f(x_i) \right],$$

where $\sigma_i$ are i.i.d. Rademacher random variables taking values $\pm 1$ with equal probability. Rademacher complexity measures the expressive capacity of the function class $\mathcal{F}$: a larger value indicates a richer class that may overfit, while a smaller value suggests better generalization ability. A standard uniform generalization bound is

$$R(f) \le \hat{R}(f) + 2\hat{\mathfrak{R}}_n(\mathcal{F}) + O\left( \sqrt{\frac{\log(1/\delta)}{n}} \right),$$

holding for all $f \in \mathcal{F}$ with probability at least $1 - \delta$.

## B    REPRODUCIBILITY STATEMENT & EXPERIMENTAL DETAILS

**Hyperparameters.** In our main experiments, the maximum sequence length for representation extraction is set to 128. For Step 1, we configure the training process as follows. The dimension of the kernel transformation $(2m)$ is set to 1024. We use the Adam optimizer with an initial learning rate of 0.01. A learning rate scheduler is employed, which reduces the learning rate by a factor of 0.5 if the performance does not improve for 3 consecutive epochs (patience = 3). The model is trained with a batch size of 100 until convergence. The weights, denoted by $W$, are set as a geometric sequence with a base of 2 and exponents ranging from 0 to num_layers $-1$. For Step 2, the model is trained for 100 epochs using the Adam optimizer. The learning rate is set to a fixed value of 1e-3, and the batch size is 128. Pruned layers in the main results are shown in the following table 5. Training was conducted on an $8\times$ NVIDIA A100 server, leveraging pipeline parallelism to distribute different layers of the model across the GPUs.

We fix the two base random matrices $\mathbf{Z}$ and $\mathbf{V}$ with their vectorizations $\text{Vec}(\mathbf{Z}) \sim \mathcal{N}_{d \times m}(0, I)$ and $\text{Vec}(\mathbf{V}) \sim \mathcal{N}_{r \times m}(0, I)$ at initialization. At each forward pass we construct frequencies $\Omega = \mathbf{D}^{1/2}\mathbf{Z} + \mathbf{L}\mathbf{V}$, where $\mathbf{D}$ and $\mathbf{L}$ are learnable. As $m \to \infty$, the empirical covariance $(1/m)\Omega\Omega^\top$ concentrates around $\mathbf{\Sigma} = \mathbf{D} + \mathbf{L}\mathbf{L}^\top$. We do not resample $\mathbf{Z}, \mathbf{V}$ during training; learning occurs via $\mathbf{D}$ and $\mathbf{L}$.

**Post training.** To counter the rise in perplexity induced by kernelized layer replacement, we apply a low-rank adaptation only at the output projection, i.e., on the final hidden-to-vocabulary mapping.

Table 5: Pruning Results for LLaMA Models.

| LLM | Pruning Ratio | Pruned Layers |
|---|---|---|
| LLaMA-2-7B | 22.8% | [6,7], [10,11], [14,15], [18,19], [22,23] |
| LLaMA-2-13B | 23.4% | [3,4], [7,8], [9,10], [17,19], [27,28] |
| LLaMA-3-8B | 25.8% | [6,7], [9,10], [12,13], [15,16], [18,19] |

When the output head is tied to the input embeddings, we insert a lightweight $1 \times 1$ linear "logit adapter" before the head and place LoRA on this adapter; otherwise, LoRA is attached directly to the head. All other parameters remain frozen (including the kernel map $\varphi_\theta$, the linear maps $\{\mathbf{A}_i\}$, and the inverse mapping), so the structural simplification of KDP is preserved while the "representation→logits" distribution is re-calibrated. Training uses teacher forcing with knowledge distillation from the unpruned teacher: the objective is $\mathcal{L} = \lambda \, \mathrm{KL}(p_{\text{teacher}} \, \| \, p_{\text{student}}) + (1 - \lambda) \, \mathrm{CE}$ with $\lambda = 0.5$ and temperature $\tau = 1.0$; we find $\tau \in [1, 2]$ behaves similarly, and report $\tau = 1.0$. LoRA hyperparameters are rank $r = 8$, $\alpha = 32$, and dropout $0.05$. Optimization uses Adam (lr $1 \times 10^{-4}$, no weight decay), bf16, gradient clip 1.0, warmup 3% with cosine decay, and early stopping on validation PPL. We train for 1–2 epochs on the calibration dataset, which is reported in Section 3.1 and sequence length of 2048. After training, the LoRA weights are merged into the output head, adding no runtime overhead.

## C    RELATED WORK

**Redundancy in LLMs.** The existence of substantial redundancy within LLMs has been demonstrated through multiple perspectives. Liu et al. (2023) and Yom Din et al. (2024) reveal that successive transformer blocks in LLMs exhibit remarkable similarity in their representational outputs. Gardinazzi et al. (2025) hierarchically analyze representations by examining topological features. They observe that the early layers exhibit numerous short-lived topological structures. In contrast, structures in the middle layers become long-lived (or persistent), before short-lived structures reappear in the late layers. This indicates that the middle layers possess significant redundancy. Song et al. (2024) indicates the residual connection structure inherent in transformer architectures constrains each block to make only incremental contributions to the overall representation, resulting in high similarity between block representations. Men et al. (2024) and Gromov et al. (2025) find multiple layers perform similar or overlapping computations to refine predictions. Lad et al. (2024) also shows that LLMs exhibit inherent redundancy, particularly in intermediate layers, where dropping middle layers yields minimal impact on final model performance. To reduce model redundancy, pruning has emerged as a mainstream method for removing parameters and improving efficiency.

**Weight Pruning.** Weight pruning is divided into structured pruning and unstructured pruning from the perspective of the pruning paradigm. Unstructured pruning removes individual weights or neurons based on predefined importance scores (Lee et al., 2019; Frantar & Alistarh, 2023; Sun et al., 2024). Although effective in reducing model size, unstructured pruning typically leads to irregular sparsity patterns, which significantly hinder hardware efficiency and deployment flexibility. Structured pruning has emerged as a powerful technique for optimizing neural networks by removing entire groups of weights while maintaining computational efficiency.

Structured pruning eliminates entire columns within weight matrices (An et al., 2024; Ashkboos et al., 2024; Le et al., 2025; Liang et al., 2025), attention heads (He et al., 2024), or even graph-based structures (Ma et al., 2023), resulting in more hardware-friendly sparsity. Despite their structural advantages, these methods often disrupt the model's original architectural flow and typically require customized software or hardware to realize actual inference acceleration.

**Layer Pruning.** To preserve architectural integrity, layer-wise structured pruning removes entire layers based on their representational redundancy. LaCo (Yang et al., 2024) groups consecutive layers and replaces them with layer-wise parameter difference to achieve compression. ShortGPT (Men et al., 2024) computes BI scores—akin to cosine similarity—between layers to identify and remove less important ones. SLEB (Song et al., 2024) estimates layer importance using perplexity and prunes those deemed insignificant. Ding et al. (2025) using Centered Kernel Alignment (CKA, Kornblith et al., 2019), map representations to a kernel space to measure similarity and remove blocks with high CKA scores. Streamlining (Chen et al., 2025) adopts cosine similarity to identify redun-

dant layers and utilizes knowledge distillation to restore performance post-pruning. However, these studies are typically metric-driven, focusing on identifying layers to prune rather than approaching the problem from a model dynamics perspective by fundamentally replacing parameter-heavy blocks with simpler modules. As a result, they either directly eliminate redundant layers (Song et al., 2024; Men et al., 2024)—inevitably incurring performance degradation—or require extensive data for fine-tuning (Kim et al., 2024; Yang et al., 2024; Gromov et al., 2025) or distillation (Chen et al., 2025) to mitigate the loss. This overlooks the opportunity to uncover simplified structural alternatives that enable rapid post-pruning recovery with minimal data.

## D PROOF

*Proof of Theorem 1.* First, we establish a high-probability generalization bound on the one-step linear approximation error. For any given step $l$, we approximate $\varphi(\mathbf{h}_{l+1})$ using a linear transformation $\mathbf{A}_l$ on $\varphi(\mathbf{h}_l)$. The error vector for this one-step prediction is defined as:

$$\mathbf{e}_l = \varphi\left(\mathbf{h}_{l+1}\right) - \mathbf{A}_l \varphi\left(\mathbf{h}_l\right). \tag{7}$$

The corresponding expected loss under the true distribution is given by:

$$L(\widehat{\theta}, \widehat{\mathbf{A}}i) = \mathbb{E}\left[\|\varphi_{\widehat{\theta}}(\mathbf{h}_{l+1}) - \widehat{\mathbf{A}}_l\varphi_{\widehat{\theta}}(\mathbf{h}_l)\|^2\right]. \tag{8}$$

We invoke the standard uniform convergence bound based on Rademacher complexity (Mohri et al., 2018, Theorem 1). Since $\|\varphi(\mathbf{h})\| \le R_\varphi$ and $\|\mathbf{A}_i\|_{\mathrm{op}} \le B_A$, the squared loss $\ell(y, y') = \|y - y'\|^2$ is $L'$-Lipschitz on the ball $\{\|y\|, \|y'\| \le (B_{\mathbf{A}}+1)R_\varphi\}$ and has range in $[0, (B_{\mathbf{A}}+1)^2 R_\varphi^2]$. Therefore, with probability at least $1 - \delta$, for all $(\theta, \{\mathbf{A}_i\})$ in the hypothesis class $\mathcal{H}' = \{\mathbf{h} \mapsto \mathbf{A}\varphi_\theta(\mathbf{h}) \in \mathbb{R}^d : \|\mathbf{A}\|_{\mathrm{op}} \le B_{\mathbf{A}}, \theta \in \Theta, \|\varphi_\theta(\mathbf{h})\| \le R_\phi\}$, it holds that

$$L(\widehat{\theta}, \{\widehat{\mathbf{A}}_i\}) \le L_{\mathrm{ERM}} + 2L'\Re_n(\mathcal{H}') + (B_{\mathbf{A}}+1)^2 R_\varphi^2 \sqrt{\frac{\log(1/\delta)}{2n}}. \tag{9}$$

Next we bound $\Re_n(\mathcal{H}')$. For each row $\mathbf{A}_l[i]$ of $\mathbf{A}_l$ we have $\|\mathbf{A}_l[i]\|^2 \le B_{\mathbf{A}}$, and $\|\varphi(\mathbf{h})\| \le R_\varphi$, hence the linear class satisfies $\Re_n(\mathcal{H}') \lesssim B_{\mathbf{A}} R_\varphi \sqrt{2m/n}$. Applying the standard contraction (shrinkage) lemma for Lipschitz losses to $\ell \circ \mathcal{H}'$ yields

$$L(\widehat{\theta}, \{\widehat{\mathbf{A}}_i\}) \le L_{\mathrm{ERM}} + C B_{\mathbf{A}}^2 R_\varphi^2 \sqrt{\frac{2m \log(2m/\delta)}{n}}. \tag{10}$$

for a universal constant $C > 0$.

**From one-step to $k$-step.** Let $e_j = \varphi_{\widehat{\theta}}(\mathbf{h}_{j+1}) - \widehat{\mathbf{A}}_j\varphi_{\widehat{\theta}}(\mathbf{h}_j)$ be the one-step residual. By a telescoping expansion and triangle inequality, we have

$$\varphi_{\widehat{\theta}}(\mathbf{h}_{l+k}) - \Big(\prod_{i=l}^{l+k-1} \widehat{\mathbf{A}}_i\Big)\varphi_{\widehat{\theta}}(\mathbf{h}_l) = e_{l+k-1} + \sum_{j=l}^{l+k-2}\Big(\widehat{\mathbf{A}}_{l+k-1}\cdots\widehat{\mathbf{A}}_{j+1}\Big)e_j. \tag{11}$$

From the Equation 10 it follows that for the learned model parameters, with probability $1 - \delta$ there is a single-step prediction error vector with the number of paradigms bounded in the:

$$\|e_l\| = \|\varphi_{\widehat{\theta}}(\mathbf{h}_{l+1}) - \widehat{\mathbf{A}}_l\,\varphi_{\widehat{\theta}}(\mathbf{h}_l)\|_{\mathcal{H}} \le \sqrt{L_{\mathrm{ERM}} + C B_{\mathbf{A}}^2 R_\varphi^2 \sqrt{\frac{2m \log(2m/\delta)}{n}}}. \tag{12}$$

Using the Equation 12, for any consecutive $k$ step state sequence, the cumulative error of $\mathcal{E}_k$ is defined as $\mathcal{E}_k = \varphi_{\widehat{\theta}}(\mathbf{h}_{l+k}) - \Big(\prod_{i=l}^{l+k-1} \widehat{\mathbf{A}}_i\Big)\varphi_{\widehat{\theta}}(\mathbf{h}_l)$. By submultiplicativity, we have:

$$\|\mathcal{E}_k\| \le \sum_{t=l}^{l+k-1}\left\|\prod_{i=t+1}^{l+k-1} \widehat{\mathbf{A}}_i\right\|_{\mathrm{op}} \|e_t\| \le \sum_{t=l}^{l+k-1} B_{\mathbf{A}}^{l+k-1-t} \|e_t\|. \tag{13}$$

Let $j = t - l \in \{0, \cdots, k-1\}$, we have:

$$\|\mathcal{E}_k\| \le \sum_{j=0}^{k-1} B_{\mathbf{A}}^{k-1-j} \|e_{l+j}\|, \tag{14}$$

here, let

$$\prod_{i=a}^{b} \widehat{\mathbf{A}}_i := \begin{cases} \widehat{\mathbf{A}}_b \widehat{\mathbf{A}}_{b-1} \cdots \widehat{\mathbf{A}}_a, & a \le b, \\ \mathbf{I}, & a > b \end{cases}. \tag{15}$$

Given that the Equation 12 has indicated, with the probability $1 - \delta$ event, each step is $\|e_j\| \le \sqrt{L_{\text{ERM}} + CB_{\mathbf{A}}^2 R_\varphi^2 \sqrt{\frac{d \log(d/\delta)}{n}}}$, we can adjust $\delta$ so that the probability applies equally to all $1 \le j \le k$ steps by means of the union bound. Therefore, under the same high probability event, substitute the above equation and extract the common factor outside the sum equation to obtain the upper bound of the cumulative error norm of the $k$ step:

$$\|\mathcal{E}_k\| \le \left( \sqrt{L_{\text{ERM}} + CB_{\mathbf{A}}^2 R_\varphi^2 \sqrt{\frac{2m \log(2m/\delta)}{n}}} \right) \cdot \left( \sum_{j=0}^{k-1} B_{\mathbf{A}}^{k-1-j} \right). \tag{16}$$

Here, $\sum_{j=0}^{k-1} B_{\mathbf{A}}^{k-1-j}$ is the linear mapping error of the $k$ step-by-step accumulation amplified coefficient via the gradual transfer. When $B_{\mathbf{A}} < 1$, the coefficient converges to $(1 - B_{\mathbf{A}})^{-1}$, and $B_{\mathbf{A}} = 1$ degenerates into a linear growth of $k$. $\qquad \square$

*Proof of Lemma 1.* We prove in two steps. First, in the ideal (infinite-dimensional) feature space induced by a universal, translation-invariant kernel (e.g., Gaussian RBF), any continuous target mapping can be approximated arbitrarily well by a linear operator on the canonical feature map. Second, by Random Fourier Features, the infinite-dimensional feature map can be approximated by a finite feature map $\varphi$ to arbitrary accuracy on the support of $\mathcal{D}$. Combining the two approximations and using the triangle inequality yields the desired bound.

**Universality of the kernel and linear representability in the RKHS feature space.** Let $\mathrm{k}(x,y)$ be a continuous, positive definite, translation-invariant kernel on $\mathbb{R}^d$ whose spectral measure has full support (e.g., the Gaussian RBF kernel). Result from (Micchelli et al., 2006) says that its RKHS $\mathcal{H}_{\mathrm{k}}$ in the tight set $\mathcal{K}$ on which is consistently dense for continuous functions: for any continuous function $f : \mathcal{K} \to \mathbb{R}$ and any $\eta > 0$, exist $f^* \in \mathcal{H}_{\mathrm{k}}$ such that $\sup_{x \subset K} |f^\star(x) - f(x)| < \eta$. Denote $\Phi : \mathbb{R}^d \to \mathcal{H}_k$ is the feature mapping, $\langle \Phi(x), \Phi(y) \rangle_{\mathcal{H}_k} = \mathrm{k}(x,y)$. Let the true mapping between layers be $\mathbf{h}_{l+1} = f(\mathbf{h}_l)$. For any $u \in \mathcal{H}_k$, consider $g_u(\mathbf{h}_l) := \langle u, \Phi(\mathbf{h}_{l+1}) \rangle_{\mathcal{H}_k} = \langle u, \Phi(f(\mathbf{h}_l)) \rangle_{\mathcal{H}_k}, u \in \mathcal{H}_k$. When $f, \Phi$ is continuous, $g_u$ is continuous in $K$. By the consistency densities in the previous paragraph, given an arbitrary $\delta > 0$, exists $f_u^* \in \mathcal{H}_k$ make

$$\sup_{h_l \in K} \|g_u(\mathbf{h}_l) - f_u^\star(\mathbf{h}_l)\| < \delta. \tag{17}$$

The nature of the regenerating nucleus suggests that $f_u^\star(\mathbf{h}_l) = \langle w_u, \Phi(\mathbf{h}_l) \rangle_{\mathcal{H}_k}$. Take a set of $\{u_j\}_{j=1}^J \subset \mathcal{H}_k$ to fetch $\phi(\cdot)$ of the former $j$ coordinates, we can get the corresponding $w_j$, and let every upper bound on the agreement error for each coordinate is $\delta_j$. Let the linear operator $\mathbf{A}_\infty \Phi(\mathbf{h}_l) := \sum_{j=1}^J \langle w_j, \Phi(\mathbf{h}_l) \rangle_{\mathcal{H}_k} u_j$. So there is a consistent error bound

$$\sup_{\mathbf{h}_l \in \mathcal{K}} \|\Phi(\mathbf{h}_{l+1}) - \mathbf{A}_\infty \Phi(\mathbf{h}_l)\|_{\mathcal{H}_k} \le \left( \sum_{j=1}^J \delta_j^2 \right)^{1/2}. \tag{18}$$

Choose $\{\delta_j\}$ so that right term $\le \varepsilon_1$ (for example, $\varepsilon_1 = \epsilon/2$), we can get

$$\sup_{\mathbf{h}_l \in \mathcal{K}} \|\Phi(\mathbf{h}_{l+1}) - \mathbf{A}_\infty \Phi(\mathbf{h}_l)\|_{\mathcal{H}_k} < \epsilon/2. \tag{19}$$

Since the expectation is less than the upper bound, there is

$$\mathbb{E}_{\mathbf{h}_l \sim \mathcal{D}} \|\Phi(\mathbf{h}_{l+1}) - \mathbf{A}_\infty \Phi(\mathbf{h}_l)\|_{\mathcal{H}_k}^2 \le (\epsilon/2)^2. \tag{20}$$

**Approximating the infinite-dimensional map by Random Fourier Features.** By Bochner's theorem (Günter, 1973), for translation-invariant kernels there exist RFF maps $\varphi : \mathbb{R}^d \to \mathbb{R}^{2m}$ such that inner products of $\varphi$ approximate the kernel. Rahimi & Recht (2007) shows that, with high probability over the draw of the features, one can uniformly approximate popular shift-invariant kernels on compact sets to within any tolerance $\varepsilon_k > 0$ using only $D = 2m = \mathcal{O}(\varepsilon_k^{-2} \log(1/\varepsilon_k))$ dimensions:

$$\sup_{x,y \in \mathcal{K}} |\langle \varphi(x), \varphi(y) \rangle - \mathrm{k}(x,y)| \le \varepsilon_k, \tag{21}$$

for any compact $\mathcal{K} \subset \mathbb{R}^d$ containing the support of the data, provided $m$ is sufficiently large (high-probability statement).

Define the (possibly infinite-dimensional) feature map $\Phi$ associated with $k$ so that $\mathrm{k}(x, y) = \langle \Phi(x), \Phi(y) \rangle$. The uniform approximation above implies that, on $\mathcal{K}$,

$$\| \langle \varphi(x), \varphi(y) \rangle - \langle \Phi(x), \Phi(y) \rangle \| \leq \varepsilon_k \quad \text{for all } x, y \in \mathcal{K}, \tag{22}$$

i.e., $x \mapsto \varphi(x)$ is an $\varepsilon_k$-isometric embedding of $x \mapsto \Phi(x)$ at the level of pairwise inner products. Let $\mathbf{A}_\infty$ be the operator satisfying

$$\sup_{h_l \in \mathcal{K}} \| \Phi(\mathbf{h}_{l+1}) - \mathbf{A}_\infty \Phi(\mathbf{h}_l) \| \leq \epsilon_1 \quad \text{with } \epsilon_1 < \frac{\epsilon}{2}. \tag{23}$$

Consider the finite-dimensional least-squares problem in RFF space and let $\mathbf{A}$ be a minimizer of

$$\min_{\| \mathbf{A} \| \leq B_\mathbf{A}} \mathbb{E}_{(h_l, h_{l+1}) \sim \mathcal{D}} \| \varphi(\mathbf{h}_{l+1}) - \mathbf{A} \varphi(\mathbf{h}_l) \|^2. \tag{24}$$

Because the squared loss expands into inner products and norms, and because RFF inner products uniformly approximate kernel inner products on $\mathcal{K}$, the objective in the RFF space uniformly approximates the kernel-space objective up to an additive $\mathcal{O}(\varepsilon_k)$ term (with constant depending only on $R_\Phi$ and $B_\mathbf{A}$. Consequently, there exists a matrix $\mathbf{A}$ with $\| \mathbf{A} \| \leq B_\mathbf{A}$ such that

$$\mathbb{E} \| \varphi(\mathbf{h}_{l+1}) - \mathbf{A} \varphi(\mathbf{h}_l) \|^2 \leq \mathbb{E} \| \Phi(\mathbf{h}_{l+1}) - \mathbf{A}_\infty \Phi(\mathbf{h}_l) \|^2 + C\varepsilon_k. \tag{25}$$

Choosing $m$ large enough so that $\varepsilon_k \leq \epsilon/(2C)$ and using the bound in Equation 20, we obtain

$$\mathbb{E}_{\mathbf{h}_l \sim \mathcal{D}} \| \varphi(\mathbf{h}_{l+1}) - \mathbf{A} \varphi(\mathbf{h}_l) \|^2 < \epsilon, \tag{26}$$

which completes this part. $\qquad\square$

*Proof of Theorem 2.* We start from Lemma 1, which guarantees the existence of an RFF map $\varphi_{\theta^*}$ (with sufficiently large $2m$) and a linear operator $\mathbf{A}^*$ such that $\mathcal{R}_{\mathrm{RFF}}(\theta^*, \mathbf{A}^*) \leq \epsilon = \mathrm{apx}_{\mathrm{ID}} - \Delta$. The rest is a finite-sample comparison: let $\widehat{\mathbf{A}}_{\mathrm{ID}}$ and $(\widehat{\theta}, \widehat{\mathbf{A}}_{\mathrm{RFF}})$ be the ERM solutions in the two classes; standard uniform-convergence bounds control their true risks by the respective optimal risks plus generalization terms $E_{\mathrm{ID}}(n, \delta)$ and $E_{\mathrm{RFF}}(n, \delta)$. Choosing $n$ so that $E_{\mathrm{RFF}}(n, \delta) + E_{\mathrm{ID}}(n, \delta) \leq \Delta$ then yields $\mathcal{R}_{\mathrm{RFF}}(\widehat{\theta}, \widehat{\mathbf{A}}_{\mathrm{RFF}}) \leq \mathcal{R}_{\mathrm{ID}}(\widehat{\mathbf{A}}_{\mathrm{ID}})$, establishing the kernel advantage with probability at least $1 - \delta$.

**Existence of a low-risk RFF predictor.** By Lemma 1, pick $\epsilon = \mathrm{apx}_{\mathrm{ID}} - \Delta > 0$. Then there exist $\theta^*$ and $\mathbf{A}^*$ with $\| \mathbf{A}^* \|_{\mathrm{op}} \leq B_\mathbf{A}$ and sufficiently large $2m$ such that $\mathcal{R}_{\mathrm{RFF}}(\theta^*, \mathbf{A}^*) \leq \epsilon$; equivalently, $\mathrm{apx}_{\mathrm{RFF}} \leq \epsilon < \mathrm{apx}_{\mathrm{ID}}$.

**Uniform convergence and ERM risks.** We now turn to finite samples. Let $\widehat{\mathbf{A}}_{\mathrm{ID}} \in \arg\min_{\mathbf{A} \subset \mathcal{F}_{\mathrm{ID}}} \widehat{\mathcal{R}}_{\mathrm{ID}}(\mathbf{A})$ and $(\widehat{\theta}, \widehat{\mathbf{A}}_{\mathrm{RFF}} \in \arg\min_{(\theta, \mathbf{A}) \in \mathcal{F}_{\mathrm{RFF}}} \widehat{\mathcal{R}}_{\mathrm{RFF}}(\theta, \mathbf{A})$. Let $\mathbf{A}_{\mathrm{ID}}^* \in \arg\min_{\mathcal{F}_{\mathrm{ID}}} \mathcal{R}_{\mathrm{ID}}(\mathbf{A})$ and $(\theta^*, \mathbf{A}^*) \in \arg\min_{\mathcal{F}_{\mathrm{RFF}}} \mathcal{R}_{\mathrm{RFF}}(\theta, \mathbf{A})$, which implies that:

$$\mathrm{apx}_{\mathrm{ID}} := \mathcal{R}_{\mathrm{ID}}(\mathbf{A}_{\mathrm{ID}}^*), \quad \mathrm{apx}_{\mathrm{RFF}} := \mathcal{R}_{\mathrm{RFF}}(\theta^*, \mathbf{A}^*). \tag{27}$$

Fix $\delta \in (0, 1)$. There exists an event $\mathscr{E}$ with probability at least $1 - \frac{\delta}{2}$ on which the following double-sided uniform deviations hold simultaneously for both classes:

$$\sup_{\mathbf{A} \subset \mathcal{F}_{\mathrm{ID}}} \left| \mathcal{R}_{\mathrm{ID}}(\mathbf{A}) - \widehat{\mathcal{R}}_{\mathrm{ID}}(\mathbf{A}) \right| \leq \varepsilon_{\mathrm{ID}}\left( n, \frac{\delta}{2} \right),$$
$$\sup_{(\theta, \mathbf{A}) \subset \mathcal{F}_{\mathrm{RFF}}} \left| \mathcal{R}_{\mathrm{RFF}}(\theta, \mathbf{A}) - \widehat{\mathcal{R}}_{\mathrm{RFF}}(\theta, \mathbf{A}) \right| \leq \varepsilon_{\mathrm{RFF}}\left( n, \frac{\delta}{2} \right), \tag{28}$$

with

$$\varepsilon_{\mathrm{ID}}(n, \delta) = C B_\mathbf{A}^2 R_h^2 \sqrt{\frac{d \log\left(\frac{cd}{\delta}\right)}{n}}, \quad \varepsilon_{\mathrm{RFF}}(n, \delta) = C B_\mathbf{A}^2 R_\varphi^2 \sqrt{\frac{2m \log\left(\frac{cm}{\delta}\right)}{n}}. \tag{29}$$

Define for brevity

$$E_{\mathrm{ID}}(n, \delta) := 2\varepsilon_{\mathrm{ID}}\left( n, \frac{\delta}{2} \right), \qquad E_{\mathrm{RFF}}(n, \delta) := 2\varepsilon_{\mathrm{RFF}}\left( n, \frac{\delta}{2} \right). \tag{30}$$

On $\mathscr{E}$, by Equation 28, we have

$$\mathcal{R}_{\mathrm{RFF}}\left(\hat{\theta}, \widehat{\mathbf{A}}_{\mathrm{RFF}}\right) \leq \mathrm{apx}_{\mathrm{RFF}} + 2\varepsilon_{\mathrm{RFF}}\left(n, \frac{\delta}{2}\right),$$

$$\mathcal{R}_{\mathrm{RFF}}\left(\hat{\theta}, \widehat{\mathbf{A}}_{\mathrm{RFF}}\right) \leq \mathrm{apx}_{\mathrm{RFF}} + E_{\mathrm{RFF}}(n, \delta). \tag{31}$$

On the same high–probability event on which the double–sided uniform deviation bounds hold, we also have

$$\mathcal{R}_{\mathrm{ID}}(\widehat{\mathbf{A}}_{\mathrm{ID}}) \geq \mathrm{apx}_{\mathrm{ID}} - E_{\mathrm{ID}}(n, \delta), \qquad E_{\mathrm{ID}}(n, \delta) := 2\,\varepsilon_{\mathrm{ID}}\left(n, \frac{\delta}{2}\right). \tag{32}$$

*Proof of Equation 32.* By the deviation bound, $\mathcal{R}_{\mathrm{ID}}(\mathbf{A}) \geq \widehat{\mathcal{R}}_{\mathrm{ID}}(\mathbf{A}) - \varepsilon_{\mathrm{ID}}(n, \delta/2)$ for all $\mathbf{A} \in \mathcal{F}_{\mathrm{ID}}$. Hence

$$\mathcal{R}_{\mathrm{ID}}(\widehat{\mathbf{A}}_{\mathrm{ID}}) \geq \widehat{\mathcal{R}}_{\mathrm{ID}}(\widehat{\mathbf{A}}_{\mathrm{ID}}) - \varepsilon_{\mathrm{ID}}\left(n, \tfrac{\delta}{2}\right) \geq \widehat{\mathcal{R}}_{\mathrm{ID}}(\mathbf{A}_{\mathrm{ID}}^{*}) - \varepsilon_{\mathrm{ID}}\left(n, \tfrac{\delta}{2}\right) \geq \mathcal{R}_{\mathrm{ID}}(\mathbf{A}_{\mathrm{ID}}^{*}) - 2\,\varepsilon_{\mathrm{ID}}\left(n, \tfrac{\delta}{2}\right),$$

where the middle inequality uses ERM optimality of $\widehat{\mathbf{A}}_{\mathrm{ID}}$ and the last one uses the deviation bound again. Since $\mathrm{apx}_{\mathrm{ID}} = \mathcal{R}_{\mathrm{ID}}(\mathbf{A}_{\mathrm{ID}}^{*})$, Equation 32 follows. □

**Comparison via the approximation margin.** From Lemma 1, there exists $\Delta > 0$ such that

$$\mathrm{apx}_{\mathrm{RFF}} \leq \mathrm{apx}_{\mathrm{ID}} - \Delta. \tag{33}$$

Combining Equation 33 with the already established bound in Equation 35 and the ID lower bound in Equation 31, we obtain on the same event:

$$\mathcal{R}_{\mathrm{RFF}}(\hat{\theta}, \widehat{\mathbf{A}}_{\mathrm{RFF}}) \leq \mathrm{apx}_{\mathrm{ID}} - \Delta + E_{\mathrm{RFF}}(n, \delta), \qquad \mathcal{R}_{\mathrm{ID}}(\widehat{\mathbf{A}}_{\mathrm{ID}}) \geq \mathrm{apx}_{\mathrm{ID}} - E_{\mathrm{ID}}(n, \delta). \tag{34}$$

Therefore, a sufficient condition for

$$\mathcal{R}_{\mathrm{RFF}}(\hat{\theta}, \widehat{\mathbf{A}}_{\mathrm{RFF}}) \leq \mathcal{R}_{\mathrm{ID}}(\widehat{\mathbf{A}}_{\mathrm{ID}})$$

is

$$E_{\mathrm{RFF}}(n, \delta) + E_{\mathrm{ID}}(n, \delta) \leq \Delta. \tag{35}$$

**Choice of Sample Size $n$.** We now choose sufficiently large $n$ to ensure that the RFF model's true risk falls below that of the identity model by a margin of at least $\Delta$.

Here we require $n$ such that the sum of the two classes' generalization error bounds is bounded by $\Delta$. Since $E_{\mathrm{RFF}}$ is typically larger than $E_{\mathrm{ID}}$, we can further simplify this to the (conservative) requirement $E_{\mathrm{RFF}}(n, \delta) \geq \Delta/2$ and $E_{\mathrm{ID}}(n, \delta) \leq \Delta/2$. Apparently it's similar the other way around. Substituting the definitions of $E_{\mathrm{RFF}}$ and $E_{\mathrm{ID}}$, we obtain the explicit condition on $n$:

$$CB_{\mathbf{A}}^2 R_\varphi^2 \sqrt{\frac{2m \log \frac{4cm}{\delta}}{n}} \leq \frac{\Delta}{4}, \quad \text{and} \quad CB_{\mathbf{A}}^2 R_h^2 \sqrt{\frac{d \log \frac{2cd}{\delta}}{n}} \leq \frac{\Delta}{4}. \tag{36}$$

This condition can be rewritten more compactly as:

$$n \geq N^*(\Delta, d, 2m, \delta) := \frac{16 \left( CB_{\mathbf{A}}^2 \left( R_\varphi^2 \sqrt{2m \log \frac{4cm}{\delta}} - R_h^2 \sqrt{d \log \frac{2cd}{\delta}} \right)_+ \right)^2}{\Delta^2}, \tag{37}$$

Here $(x)_+ = \max\{x, 0\}$ denotes the positive part, which appears because if the term in parentheses is negative (i.e. if $R\varphi^2 \sqrt{2m \log(4cm/\delta)} < R_h^2 \sqrt{d \log(2cd/\delta)}$), then the RFF class actually has no larger complexity term than the ID class, and the sample requirement can be significantly weak. In the typical case $2m \gg d$, the term in parentheses will be positive, and the $N^*$ given above is the dominant sample complexity threshold. □

# E  EXPERIMENTS ON OPT MODEL

The performance is further evaluated on two OPT models (OPT-2.7B and OPT-6.7B, Zhang et al., 2022) with the results presented in Table 6 and Table 7. The results indicate that our method maintains robust performance even on relatively small models. While experiencing a marginal decline on generation tasks, our method's performance on generation tasks stays comparable, showing no signs of collapse.

Table 6: Performance comparison on OPT-2.7B and OPT-6.7B.

| LLM | Method | CMNLI | HeSw | PIQA | CHID | WSC | CoQA | BoolQ | MMLU | CMMLU | Race | SST2 | C3 | Avg. | RP. |
|---|---|---|---|---|---|---|---|---|---|---|---|---|---|---|---|
| OPT-2.7B | Dense | 34.2 | 45.2 | 75.4 | 37.9 | 78.0 | 51.1 | 60.6 | 25.9 | 26.3 | 35.6 | 54.3 | 36.8 | 46.78 | 100.0 |
| | SLEB | 30.4 | 40.4 | 74.2 | 22.1 | 72.5 | 44.8 | 58.2 | 25.0 | 25.3 | 31.0 | 55.4 | 35.6 | 42.91 | 91.7 |
| | SliceGPT | 31.0 | 27.8 | 50.5 | 19.9 | 57.1 | 37.3 | 50.7 | 24.8 | 23.4 | 25.8 | 60.6 | 25.1 | 36.17 | 77.3 |
| | SliceGPT[†] | 33.6 | 40.8 | 63.1 | 24.1 | 58.5 | 37.0 | 46.5 | 25.1 | 24.5 | 25.0 | 53.4 | 26.0 | 38.12 | 81.5 |
| | w/o Kernel | 30.8 | 34.5 | 57.1 | 15.8 | 69.1 | 11.2 | 50.3 | 25.3 | 24.8 | 24.5 | 51.5 | 23.5 | 34.87 | 74.5 |
| | Ours | 31.9 | 37.9 | 73.5 | 33.1 | 72.7 | 30.7 | 59.4 | 25.7 | 25.7 | 33.2 | 61.0 | 35.4 | 43.35 | 92.6 |
| OPT-6.7B | Dense | 33.4 | 60.2 | 78.6 | 42.5 | 81.7 | 55.0 | 63.2 | 25.3 | 25.5 | 35.3 | 76.8 | 39.6 | 51.43 | 100.0 |
| | SLEB | 36.2 | 53.2 | 76.6 | 23.1 | 74.7 | 43.7 | 46.2 | 24.1 | 24.8 | 31.0 | 71.6 | 38.6 | 45.32 | 88.1 |
| | SliceGPT | 30.3 | 27.0 | 62.1 | 18.8 | 63.4 | 21.8 | 33.3 | 24.8 | 25.3 | 24.0 | 55.7 | 26.5 | 34.42 | 66.9 |
| | SliceGPT[†] | 30.0 | 29.3 | 61.7 | 18.8 | 69.5 | 27.6 | 34.7 | 29.7 | 25.0 | 27.1 | 62.0 | 37.7 | 37.76 | 73.4 |
| | w/o Kernel | 32.8 | 63.1 | 67.5 | 23.6 | 72.5 | 39.3 | 40.6 | 25.0 | 24.2 | 23.3 | 61.7 | 33.5 | 42.26 | 82.2 |
| | Ours | 33.7 | 53.0 | 74.5 | 27.1 | 76.5 | 44.5 | 49.4 | 29.3 | 26.1 | 31.3 | 69.0 | 39.5 | 46.16 | 89.8 |

Table 7: Performance of OPT model on generation benchmarks. For Perplexity, a lower score means better performance ($\downarrow$), while for ROUGE, a higher score means better performance ($\uparrow$).

| LLM | Method | WIKI$\downarrow$ | C4$\downarrow$ | Xsum$\uparrow$ | RP(PPL) | RP(Rouge) |
|---|---|---|---|---|---|---|
| OPT-2.7B | Dense | 15.1 | 31.3 | 6.5 | 100.0 | 100.0 |
| | Ours | 28.0 | 37.2 | 2.8 | 69.0 | 43.1 |
| OPT-6.7B | Dense | 10.9 | 12.3 | 13.7 | 100.0 | 100.0 |
| | Ours | 13.9 | 22.1 | 6.0 | 67.0 | 43.8 |

# F MORE ABLATION STUDIES

**Different Pruning Ratios(HellaSwag and BoolQ).** Although CMMLU (Figure 7) is a comprehensive benchmark, the dense model's performance on it remains relatively low. To mitigate this potential side effect, we present performance curves on HellaSwag and BoolQ across various pruning rates. The results confirm that the overall performance still exhibits a consistent linear decline.

**Different Kernel Sizes.** A kernel function for our objective must address a fundamental trade-off; it must be powerful enough to simplify the system's dynamics, yet sufficiently constrained to retain the core information required by the task. For example, a trivial kernel that maps all representations to a constant would perfectly linearize the dynamics, but at the cost of a complete "information collapse" that invalidates the model. Therefore, our goal is to design a non-trivial kernel that can effectively model the simplified dynamics on the slow manifold. Given such a kernel, we can simplify the transformations between layers, thereby enabling layer pruning. To validate this point, beyond the end-to-end downstream experiments in the main text, we further investigate how the method's performance varies with different kernel sizes (i.e., $2m$).

Figure 8 illustrates the impact of varying kernel sizes on the performance of LLaMA2-7B across three representative datasets (Hellaswag, CoQA and MMLU). With smaller kernel sizes, such as 256

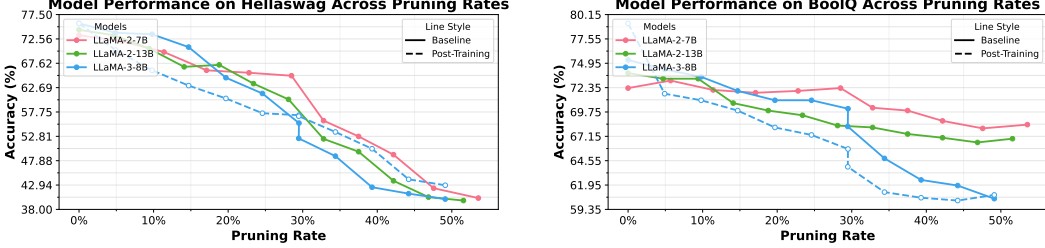

Figure 7: Performance of HellaSwag and BoolQ when we prune with increasing pruning ratio.

and 512, there is a significant drop in performance, which suggests that critical information is lost due to excessive compression. Conversely, once the kernel size reaches 1024, the performance gains become marginal. This indicates that a 1024-dimensional kernel space, induced by our learnable RFF can encode the model's essential information.

**Data efficiency.** As presented in Table 8, our method demonstrates high data efficiency, utilizing only marginally more data than SliceGPT. Critically, our method does not require any post-training to restore performance.

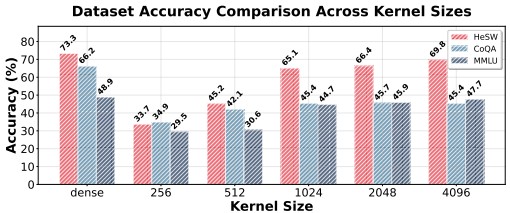

Figure 8: Ablation study on kernel size $m$.

Table 8: Data usage statistics for each methods.

| Method | Need Training | Need Fine-tuning | Total Sample |
|---|---|---|---|
| SLEB | ✗ | ✗ | / |
| ShortGPT | ✗ | ✗ | / |
| LaCo | ✗ | ✔ | 1B |
| Streamline | ✔ | ✔ | 30k |
| LLMPruner | ✔ | ✔ | 50k |
| SliceGPT | ✔ | ✗ | 1k |
| Ours | ✔ | ✗ | 4k |

**Sensitivity analysis for each step.** To further investigate the importance of each step, we conduct an ablation study under extreme conditions on LLaMA-2-7B. We individually train either Step 1 or Step 2 for a minimal number of epochs, forcing the subsequent steps to handle all performance modeling. As shown in Table 9, we observe a significant performance collapse in both scenarios. A significant discrepancy in performance becomes evident after a short training period. Training Step 1 for just 5 epochs, despite recovering some performance, still falls considerably short of the original baseline. Conversely, training Step 2 for five epochs achieves performance almost on par with the fully trained method. This provides strong support for our central thesis presented in the main text: that the trained kernel function models performance, while the inverse transformation network's role is to learn the corresponding inverse transformation.

Table 9: Sensitivity analysis for each step.

| Method | HeSW | CoQA | MMLU |
|---|---|---|---|
| Original | 65.1 | 45.4 | 28.1 |
| Barely Step 1 (1 Epoch) | 43.9 | 17.0 | 24.2 |
| Barely Step 1 (5 Epoch) | 52.1 | 27.0 | 26.7 |
| Barely Step 2 (1 Epoch) | 43.7 | 13.0 | 24.4 |
| Barely Step 2 (5 Epoch) | 60.0 | 43.7 | 26.3 |

**Speed up.** To further assess the practicality of our method, we compare its performance and efficiency across several mainstream LLaMA models, with detailed results in Table 10. We mainly focus on focuses on quantifying the impact of reducing inference computation (FLOPs) on the models' average performance (Avg RP). For LLaMA-2-7B, FLOPs were reduced by 23.13%, which brought a 1.30x theoretical inference speedup. For LLaMA-2-13B, FLOPs were reduced by 23.64%, achieving a 1.30x speedup, and the performance retention rate reached 92.9%; this is the result with the best performance-efficiency trade-off among the three. In summary, the experimental results demonstrate that our method effectively accelerates the inference of LLMs. It offers a tunable mechanism for trading acceptable performance degradation for significant gains in computational efficiency. This feature holds substantial practical value for deploying large models in resource-constrained environments.

**Different training datasets.** To evaluate the performance of our method across various datasets, we conduct experiments on the LLaMA2-7b model. The results are presented in Table 11. Specifically, the "Fewer" setting utilizes a training set comprising 200 samples each from BoolQ, CMMLU, and Slimpajama. In contrast, the "More" setting augmented this base dataset with data from CoQA and HellaSwag, totaling 11.4k samples. Here, "Avg." denotes the average performance across the datasets. "RP. on Dense" and "RP. on std." indicate the Retained Performance relative to the Dense model and the standard dataset (reported in the main text), respectively. The experimental results

Table 10: Performance and efficiency comparison.

| Model | Method | FLOPs (G) | Change (%) | SpeedUp |
|---|---|---|---|---|
| LLaMA-2-7B | Dense | 845.71 | – | – |
| | SLEB | 664.38 | 27.29 | 1.27× |
| | ShortGPT | 612.57 | 38.06 | 1.38× |
| | Streamline | 586.67 | 30.63 | 1.44× |
| | SliceGPT | 717.63 | 15.14 | 1.17× |
| | **Ours** | 650.06 | 23.13 | 1.30× |
| LLaMA-2-13B | Dense | 1645.01 | – | – |
| | SLEB | 1320.20 | 19.74 | 1.24× |
| | ShortGPT | 1239.00 | 24.68 | 1.33× |
| | Streamline | 1229.76 | 25.24 | 1.34× |
| | SliceGPT | 1392.89 | 15.32 | 1.18× |
| | **Ours** | 1255.99 | 23.64 | 1.30× |

demonstrate the remarkable efficiency of our method. Our standard configuration ("Standard"), using only 4k data samples, achieved 91.4% of the performance of the Dense model (average score of 58.69). Notably, the "Fewer" configuration, with merely 0.6k samples, attained 97.7% of the performance of the standard configuration, indicating that our method is not sensitive to the data volume and possesses strong generalization capabilities. Furthermore, by increasing the dataset size, the "More" configuration surpassed the standard version's performance (103.1%), which validates the scalability of our method. In conclusion, this ablation study highlights the robustness of our method with respect to data quantity.

Table 11: Performance comparison of LLaMA-2-7B under different training data setups.

| LLM | Datasets | CMNLI | HeSw | PIQA | CoQA | BoolQ | MMLU | CMMLU | Race | SST2 | C3 | Avg. | RP. on Dense | RP. on std. |
|---|---|---|---|---|---|---|---|---|---|---|---|---|---|---|
| LLaMA-2-7B | Dense | 34.9 | 73.3 | 79.7 | 66.2 | 72.3 | 48.9 | 30.8 | 42.8 | 93.2 | 44.8 | 58.69 | 100.0 | - |
| | Standard (8d 4k) | 33.6 | 65.1 | 70.1 | 45.4 | 71.6 | 44.7 | 28.1 | 39.9 | 94.0 | 43.7 | 53.62 | 91.4 | 100.0 |
| | Fewer (3d 0.6k) | 33.1 | 72.7 | 76.5 | 46.0 | 72.7 | 29.3 | 26.4 | 36.3 | 90.5 | 40.2 | 52.37 | 89.2 | 97.7 |
| | More (10d 11.4k) | 34.9 | 73.0 | 77.0 | 47.9 | 70.7 | 44.7 | 30.3 | 38.0 | 92.4 | 44.3 | 55.32 | 94.3 | 103.1 |

## G   MORE VISUALIZATION

This section presents additional visualizations of the fitting results. Following the mechanism previously detailed in Figure 4, these figures provide a clearer illustration of the fitting quality.

## H   MORE ANALYSIS OF SLOW MANIFOLD HYPOTHESIS

The manifold hypothesis posits that high-dimensional data often reside on a manifold with a dimensionality much lower than that of the ambient space, which is beneficial in various fields (Song & Ermon, 2019). This leads us to consider whether this property also holds for the LLM's internal representations, allowing them to be embedded in a low-dimensional manifold. The internal representations of the model are sequential. As shown in Equation 1, in standard Pre-Norm Transformers, $h_{l+1}$ is obtained by adding an increment to $h_l$. We therefore define this increment as the velocity of change in the representation.

Although CKA and Cosine similarity are measures of similarity rather than distance, they are closely correlated with distance measures on the manifold. This implies that high similarity between two representations corresponds to a small distance between them. This two similarity matrices in Figure 1 provide quantitative evidence for this. When the representation change over a single propagation step is small, we consider the representation to be in a stable state on the manifold, which is often feasible to model this stable evolution using a simple dynamical model.

The concept of using linear approximations for nonlinear dynamics finds parallels in many fields. Beyond the fundamental example of calculus, a typical case in partial differential equations (PDEs)

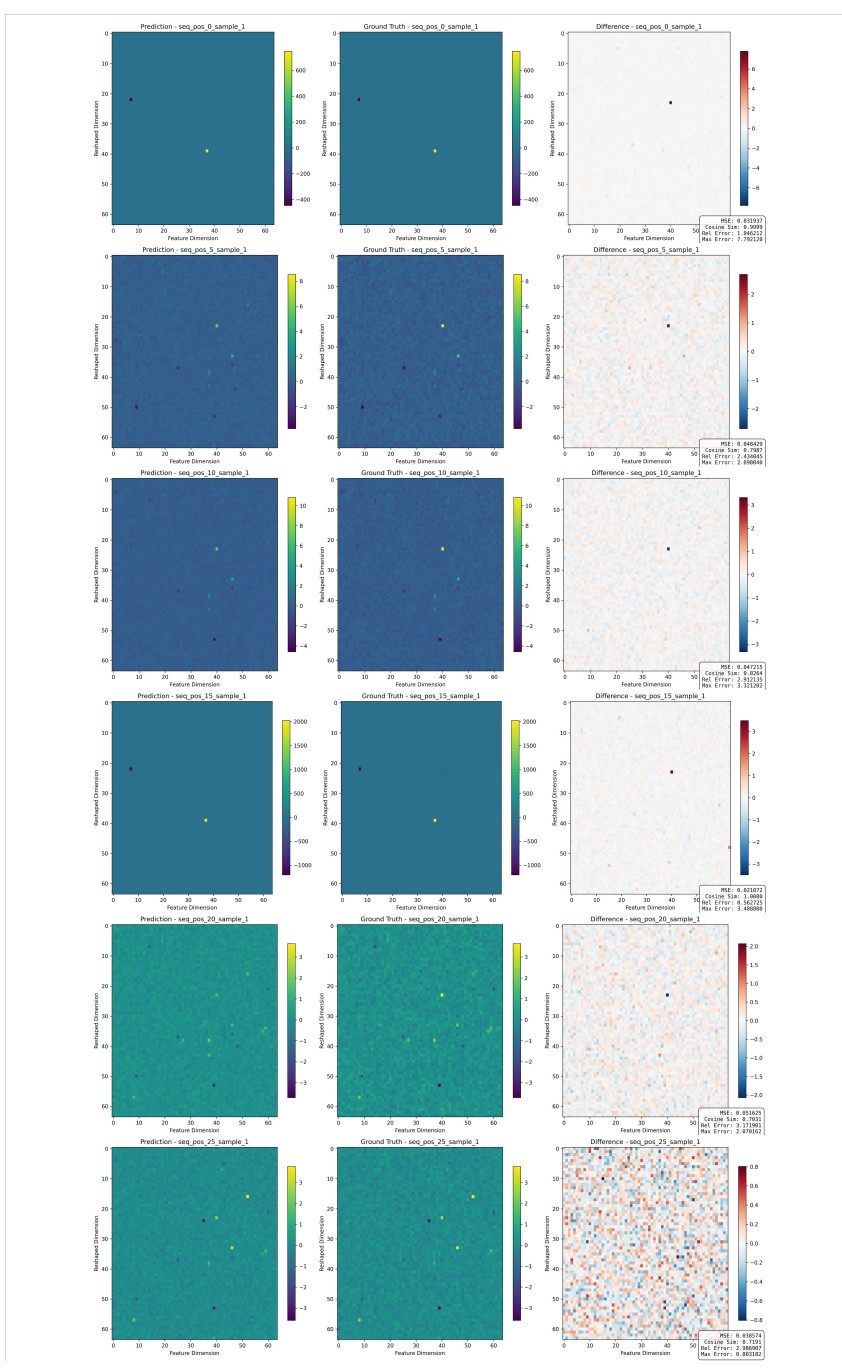

Figure 9: Additional Visualizations of Representation Fitting.

Figure 10: The representation propagation trajectories for 100 samples within LLaMA2-7B. The X and Z axes denote the t-SNE coordinates, while the Y-axis represents the layer index. An inspection along the Y-axis (layer depth) reveals that the trajectories for individual samples remain well-separated and exhibit strong linearity. Furthermore, the transformation between adjacent layers is gradual.

is the Fisher equation. Its full form is: $u_t = Du_{xx} + ru - \frac{r}{K}u^2$. However, in the vicinity of $u = K$, this equation is often simplified into the following linear PDE: $v_t = Dv_{xx} - rv$.

To provide further intuitive support for our hypothesis, we visualized the representational trajectories across different layers for 100 samples from LLaMA2-7B using t-SNE in Figure 10. The resulting plot clearly exhibits a strong linear trend, which in turn validates the feasibility of our method.

## I   ADDITIONAL OVERHEAD ANALYSIS

KDP introduces additional parameters during two steps: Kernel Linearization Joint Training requires training the parameters $\theta$, $\mathbf{A}_i$ defined in Equation 5; Inverse Transformation Network involves training the parameter $\phi$ from Equation 6. Specifically, when we replace a block of $K_{\max}$ consecutive layers, the parameter counts for the three components ($\theta$, $\mathbf{A}_i$, $\phi$) are as follows:

$\theta$: The RFF kernel learns a data-driven anisotropic Gaussian RBF kernel. Its learnable parameters $\theta$ are derived from the parameterization of the covariance matrix $\Sigma = \mathbf{D} + \mathbf{L}\mathbf{L}^\top$. This parameterization consists of two components: First, the diagonal matrix $\mathbf{D} = \mathrm{diag}(\exp(\lambda))$, is parameterized by the learnable vector $\lambda \in \mathbb{R}^d$, contributing $d$ parameters. Second, $\mathbf{L} \in \mathbb{R}^{d \times r}$ is a learnable low-rank factor matrix, which contributes $d \times r$ parameters. Therefore, the total parameter count is: $\mathrm{Params}(\theta) = d + dr = d(1 + r)$.

$\mathbf{A}_i$: Each operator $\mathbf{A}_i$ maps the kernel space representation $\varphi(\mathbf{h}_{l+i-1}) \in \mathbb{R}^{2m}$ to $\varphi(\mathbf{h}_{l+i}) \in \mathbb{R}^{2m}$. Consequently, $\mathbf{A}_i$ is a $(2m) \times (2m)$ matrix. To ensure stable training, $\mathbf{A}_i$ is parameterized as $\mathbf{A}_i = \mathbf{I} + \gamma_i \mathbf{B}_i$. The learnable parameters for each $\mathbf{A}_i$ thus consist of the scalar $\gamma_i$ and the matrix $\mathbf{B}_i \in \mathbb{R}^{2m \times 2m}$. This results in a total parameter count of $\mathrm{Params}(\mathbf{A}_i) = K_{\max} \cdot (1 + (2m)^2)$. Notably, once training is complete, only the product $\left(\prod_{i=1}^{k} \widehat{\mathbf{A}}_i\right)$ needs to be stored. Consequently, during the inference phase, the final parameter count is reduced to only: $\mathrm{Params}(\mathbf{A}_i) = (2m)^2$.

textbf$\phi$:  This represents the parameters for the inverse transformation network $\mathcal{I}(\cdot)$. The network is parameterized as $\mathcal{I}(x) := \alpha \cdot \mathrm{MLP}(x)$, where $\alpha$ is a learnable scalar and $\mathrm{MLP}(\cdot)$ stands for a two-layer MLP network. The total parameter count is given by: $\mathrm{Params}(\phi) = 1 + (2m \cdot d_{\mathrm{hidden}} + d_{\mathrm{hidden}}) + (d_{\mathrm{hidden}} \cdot d + d)$.

Therefore, during the training phase, the total introduced parameter count is

$$\mathrm{Params}^{\mathrm{Trian}}(\mathrm{KDP}) = d(1 + r) + K_{\max}(1 + (2m)^2) + (1 + d_{\mathrm{hidden}}(2m + d + 1) + d),$$

and during the inference phase, the total introduced parameter count is

$$\text{Params}^{\text{Inference}}(\text{KDP}) = d(1 + r) + (2m)^2 + (1 + d_{\text{hidden}}(2m + d + 1) + d).$$

The FLOPS and corresponding parameter counts under different $k$ are in the Table 12, which demonstrate that the additional overhead introduced by KDP is minimal.

Table 12: FLOPs and parameter retention under $k = 2$ and $k = 3$.

| LLM | | FLOPs (G) | | Params (M) | |
|---|---|---|---|---|---|
| | | $k = 2$ | $k = 3$ | $k = 2$ | $k = 3$ |
| LLaMA-2-7B | Transformer Block | 51.80 | 77.71 | 404.77 | 607.15 |
| | Fold Block | 2.89 | | 12.40 | |
| | Retention Rate | 5.59% | 3.73% | 3.07% | 2.05% |
| LLaMA-2-13B | Transformer Block | 81.20 | 121.80 | 634.41 | 951.61 |
| | Fold Block | 3.4 | | 14.83 | |
| | Retention Rate | 4.19% | 2.79% | 2.34% | 1.56% |

## J  THE USE OF LLMS

We thank the Large Language Model (LLM) for its assistance in proofreading and polishing the manuscript's language. The LLM was not involved in the idea, theoretical development, or experimental aspects of this research.

