# OpenReview forum: "KDP: Simplifying Representation Dynamics in Kernel Space"
_ICLR.cc/2026/Conference — ICLR 2026 Poster_

### Official Review · Reviewer_D7Lt · 2025-10-28

**Soundness:** 3
**Presentation:** 3
**Contribution:** 2
**Rating:** 6
**Confidence:** 3

**Summary:**

The paper introduces Kernelized Dynamics Pruning (KDP), a new method for pruning layers in large language models by viewing their forward computation as a discrete-time dynamical system. It observes that consecutive layers often produce highly similar representations, indicating redundancy. To exploit this, KDP projects representations into a kernel space where nonlinear transformations become approximately linear, allowing simpler modeling. A linear operator and an inverse mapping network then replace entire Transformer blocks. The authors present a theoretical error bound showing that multi-layer dynamics can be linearly approximated in this kernel space and prove that it provides superior fitting capacity compared to the original representation space. Extensive experiments on fifteen benchmarks demonstrate that KDP effectively prunes models while maintaining performance, without requiring fine-tuning. Overall, KDP provides a geometric and theoretically principled framework for simplifying internal model dynamics and reducing redundancy in large language models.

**Strengths:**

- Originality: while the fact that LLMs are characterised by highly redundant layers is well known, with several papers attempting to interpret this fact in different ways, this work’s idea on exploiting kernel space to identify similarity and learn the linear map to prune layers is, to the best of my knowledge, novel and effective.
- Quality: the authors take a significant effort in stating clearly their theory and in building a solid set of experiments
- Clarity: The problem is clearly set and the results and experiments are well setup and presented.
- Significance: the effort of reducing LLM is certainly timely and important. Before this work, papers arguing that layers of pre-trained models were redundant mostly offered solutions which dealt with pruning layers and then fine-tune to cure the cut. The proposal of this paper of learning the map to bypass the pruned layers seems to be a relevant step forward.

**Weaknesses:**

I have a couple of general objections:
- About the broad framing of the paper: the authors talk frequently (abstract, introduction) about the fact that they view pre-trained models as “discrete-time dynamical systems” and talk about “slow manifold” and “reduced velocity” during the redundant phase. These concepts are not developed throughout the paper, and it would seem that the paper’s results and claims would be unaffected by removing this interpretation. While I’m totally in favour of trying to give a physical interpretation of LLMs representations, I think the paper would benefit if the authors could expand on this interpretation more (maybe in appendix?), or remove it altogether.
- I think the paper might benefit from a short interpretability discussion about which layers exactly are being cut and why, also wrt to previous literature. Previous work (ex. Men et al. as referenced in the paper) seem to show that layers after the first half of the model depth are good candidate for pruning, and that cutting earlier, or late layers, causes a drop in performance. A study of this on similar models is found e.g. on Gardinazzi et al. ICML 2025. In other words: there are two things determining viable pruning: size of the cut and position of the cut. The latter is not discussed from what I can see, and this work might have a good angle on this, starting from theorem 1. Moreover, in appendix it seems that several non-consecutive blocks are pruned, which is in contrast with previous work: is this discussed anywhere in the paper?
- The theoretical part, especially around theorem 1, suffers a bit from lack of background on the techniques used: the authors talk about ERM (empirical risk minimization?), population risk, etc without connecting this to the problem at hand. To someone not familiar with these concepts, this is a bit hard to follow.

**Questions:**

Here are a few more detailed questions that I’d like to be addressed
- From Figure 1, it seems to me that CKA saturates to 1 very frequently (particularly for llama3-8B). It is argued that this is due to similarity in kernel space being more effective. However, if I understood correctly theorem 1 argues that stacking too many layers accumulate errors, even if they are all similar. Indeed, from the results it seems that the algorithm prefers to cut non consecutive layer blocks, than one-two larger consecutive blocks, which would be more similar to previous work. Which part exactly of the algorithm deals with this decision? if it is entirely based on CKA value, and the value is very close to one across the whole depth, is this stable?
- I haven’t found a definition of ERM, first mentioned in theorem 1.
- Did the authors verify whether the algorithm would cut early and late layers and verify the drop in performance? they say they exclude them based on previous literature, but would they be excluded by the algorithm anyway or is this to be input as prior knowledge?
- Can the author explain exactly what “we prune about 25% of the parameters mean in 4.1 setup-models?
- in the ablation study, the authors use CMMLU, for which performance is low for all models even for the dense case. Wouldn’t it be safer to consider a benchmark for which the dense case performs relatively well? Testing around random performance might have some side effect.

---

> ### Author Response · Authors · 2025-11-20
> **Rebuttal Reply 1/n**
>
> We would like to express our gratitude for your careful reading and valuable comments, which substantially improved our paper. In the following, we address your concerns one by one. Your comments are shown by **W** and **Q** (**W** denotes Weakness; **Q** denotes Question) and are followed by our responses. For all page numbers and section numbers in this reply, please refer to the revised paper unless otherwise specified.
>
> >W1:  About the broad framing of the paper: the authors talk frequently (abstract, introduction) about the fact that they view pre-trained models as "discrete-time dynamical systems" and talk about "slow manifold" and "reduced velocity" during the redundant phase. These concepts are not developed throughout the paper, and it would seem that the paper's results and claims would be unaffected by removing this interpretation. While I'm totally in favour of trying to give a physical interpretation of LLMs representations, I think the paper would benefit if the authors could expand on this interpretation more (maybe in appendix?), or remove it altogether.
>
> Thanks for your advice.  Due to space constraints in the previous manuscript,  we did not provide a detailed explanation of dynamics-inspired interpretation. We adopt your advice to give further analysis, we have added these supplementary materials to the **Appendix H** and **Section 2.2 (Line 158)**.
>
> We emphasize viewing the forward propagation of neural networks as a discrete-time dynamical system because this serves as a key foundational inspiration for our method; our entire framework was subsequently constructed around this concept.
>
> Our method is inspired by the well-known "manifold hypothesis," which posits that high-dimensional data often reside on a manifold with a dimensionality much lower than that of the ambient space. This hypothesis has also proven beneficial in various fields, including diffusion models [1].
>
> This leads us to consider whether this property also holds for the model's internal representations, allowing them to be embedded in a low-dimensional manifold.  The internal representations of the model are sequential. As shown in **Equation 1**, in standard Pre-Norm Transformers, $\mathbf h_{l+1}$ is obtained by adding an increment to $\mathbf h_l$. We therefore define this increment as the velocity of change in the representation.
> Although CKA and Cosine similarity are measures of similarity rather than distance metrics, they are closely correlated with distance measures on the manifold. This implies that high similarity between two representations corresponds to a small distance between them.  The Cosine and CKA similarity matrices (**Figure 1**) provide quantitative evidence for this. The extensive bright red regions within these matrices, particularly for the CKA measure, indicate that the representations across multiple consecutive layers are highly similar, with scores approaching 1.0.  When the representation change over a single propagation step is small, we consider the representation to be in a stable state on the manifold. It is then often feasible to model this stable evolution using a simple dynamical model.
>
> The concept of using linear approximations for nonlinear dynamics finds parallels in many fields. Beyond the fundamental example of calculus, a representative example is Linear Stability Analysis, which is widely applied in partial differential equations (PDE).
>
> To provide further intuitive support for our hypothesis, we visualize the representational trajectories across different layers for 100 samples from LLaMA2-7B using t-SNE (**Figure 10**). The resulting plot clearly exhibits a strong linear trend, which in turn validates the feasibility of our method.
>
> [1] Song Y, Ermon S. Generative modeling by estimating gradients of the data distribution[J]. Advances in neural information processing systems, 2019, 32.

---

> > ### Author Response · Authors · 2025-11-20
> > **Rebuttal Reply 2/n**
> >
> > >W2: I think the paper might benefit from a short interpretability discussion about which layers exactly are being cut and why, also wrt to previous literature. Previous work (ex. Men et al. as referenced in the paper) seem to show that layers after the first half of the model depth are good candidate for pruning, and that cutting earlier, or late layers, causes a drop in performance. A study of this on similar models is found e.g. on Gardinazzi et al. ICML 2025. In other words: there are two things determining viable pruning: size of the cut and position of the cut. The latter is not discussed from what I can see, and this work might have a good angle on this, starting from theorem 1. Moreover, in appendix it seems that several non-consecutive blocks are pruned, which is in contrast with previous work: is this discussed anywhere in the paper?
> >
> > Thank you for raising this critical question regarding pruning "position" and "size".  We adopt your advice to give more statements on page 15(**Line 785**) in the revision. Regarding the "pruning location," our findings are fully consistent with those in the literature (e.g., Men et al. [1]): the middle layers are the primary source of redundancy, whereas the early and late layers are more sensitive to pruning. Indeed, we empirically observe that pruning the initial or final layers leads to substantial performance degradation. The work by Gardinazzi et al. [2] , which the reviewer noted, provides an excellent topological explanation for this. They find that the model's middle layers exhibit "long-lived topological structures," indicating that the representational geometry tends to stabilize in these layers. This aligns perfectly with our "Slow Manifold" hypothesis. Gardinazzi's "persistent topological features" and our "high CKA similarity" (**Figure 1**) are likely analyses of the same underlying phenomenon from distinct mathematical perspectives (topology vs. kernel space dynamics).
> >
> > Our method does not simply "remove" layers; rather, it "replaces" them with an approximating module. **Theorem 1** provides a clear theoretical upper bound, showing that the approximation error accumulates non-linearly with the number of consecutively replaced layers, $k$ (i.e., $K_{max}$).
> >
> > Intuitively, we posit that non-consecutive blocks allow the representation to be briefly re-aligned with the original space, a mechanism that prevents the accumulation of error. Our ablation study (**Table 4 (Line 486)**) provides strong validation for this hypothesis. As detailed in **Section 4.4**, model performance remains stable when $K_{max} \le 3$; however, as $K_{max}$ increases further, performance "catastrophically degrades". The "multiple non-consecutive blocks" pruning pattern (**Table 5**) observed by the reviewer is not coincidental; it is a core design feature of the KDP method. This pattern represents the optimal solution our algorithm finds between the competing forces of "identifying redundancy" (via CKA) and "avoiding cumulative error" (via **Theorem 1** and the $K_{max}$ constraint). It allows KDP to maximize redundancy removal (using multiple short blocks) while maintaining model stability (by preventing $K_{max}$ from becoming too large).
> >
> > [1] Xin M, Mingyu X, Qingyu Z, Bingning W et. al. ShortGPT: Layers in Large Language Models are More Redundant Than You Expect[C]. In Findings of the Association for Computational Linguistics, 2024.
> >
> > [2] Yuri G, Karthik V, Giada P et. al. Persistent topological features in large language models[C]. In International Conference on Machine Learning, 2025.

---

> > > ### Author Response · Authors · 2025-11-20
> > > **Rebuttal Reply 3/n**
> > >
> > > >W3 and Q2: The theoretical part, especially around theorem 1, suffers a bit from lack of background on the techniques used: the authors talk about ERM (empirical risk minimization?), population risk, etc without connecting this to the problem at hand. To someone not familiar with these concepts, this is a bit hard to follow. I haven't found a definition of ERM, first mentioned in theorem 1.
> > >
> > > Thank you for this valuable feedback.  We agree with you and adopt your advice to add some detailed explanations of these concepts  in **Appendix A** in the revision.
> > >
> > > >Q1:  From Figure 1, it seems to me that CKA saturates to 1 very frequently (particularly for llama3-8B). It is argued that this is due to similarity in kernel space being more effective. However, if I understood correctly theorem 1 argues that stacking too many layers accumulate errors, even if they are all similar. Indeed, from the results it seems that the algorithm prefers to cut non consecutive layer blocks, than one-two larger consecutive blocks, which would be more similar to previous work. Which part exactly of the algorithm deals with this decision? if it is entirely based on CKA value, and the value is very close to one across the whole depth, is this stable?**
> > >
> > > This is a very perceptive observation. You are entirely correct.
> > > **Theorem 1** advises against pruning an excessive number of continuous layers, as this error accumulates exponentially. We deliberately set $K_{max}$ to a very small value (e.g., 2 or 3), a decision that stems directly from the theoretical warning in **Theorem 1** and the experimental validation in **Table 4**. As described in **Algorithm 1**, we select $B$ non-overlapping blocks to achieve the desired total pruning rate. Therefore, your observation that the algorithm favors pruning multiple, non-continuous short blocks is a direct result of the algorithm strictly adhering to the $K_{max}$ constraint to proactively avoid the error accumulation that Theorem 1 warns against.
> > >
> > > However, we observe a potential negative effect when the inter-layer similarity frequently saturates at: the PPL may increase sharply with the pruning rate (as shown in **Figure 6**). (A detailed analysis is provided in our response to Q1 from Reviewer d1gK.) We are actively exploring methods to enhance stability.
> > >
> > > >Q3: Did the authors verify whether the algorithm would cut early and late layers and verify the drop in performance? they say they exclude them based on previous literature, but would they be excluded by the algorithm anyway or is this to be input as prior knowledge?
> > >
> > > Thank you for your valuable comments.
> > > Consistent with prior work, we exclude the initial and final 10% of layers from our analysis (**Line 244**). Furthermore, as specified in **Algorithm 1**, we ensure that consecutive layer blocks are non-overlapping (**Line 112**).
> > > We observe a substantial performance degradation when pruning the early (e.g., [1,2,3]) or late (e.g., [29,30,31]) layers, as opposed to pruning based on our selection criteria. We assume that this is because the initial and final layers of the large model are more specialized for input comprehension and output formulation, respectively.
> > >
> > > We conceptualize the forward propagation as a progressive transformation of a latent space. In the early and late layers, this latent space remains closer to the Token Space, which appears to be more sensitive to perturbations. We are actively pursuing this line of investigation.
> > >
> > > Notably, this finding aligns with the Cosine similarity trends depicted in **Figure 1**. This consistency prompts us to reconsider that, while the CKA-based selection criteria are effective, Cosine similarity itself may serve as a viable metric for identifying prunable layers.

---

> > > > ### Author Response · Authors · 2025-11-20
> > > > **Rebuttal Reply 4/n**
> > > >
> > > > **Q4: Can the author explain exactly what "we prune about 25% of the parameters" mean in 4.1 setup-models?**
> > > >
> > > > We have revised the corresponding text accordingly (**Line 306**). This suggests an approximate 25% reduction in the original model parameters. However, as our method also incorporates supplementary parameters after pruning, the number of layers designated for removal must be determined based on the pruning budget. The specific layers pruned are detailed in **Table 5**.
> > > >
> > > > **Q5: In the ablation study, the authors use CMMLU, for which performance is low for all models even for the dense case. Wouldn't it be safer to consider a benchmark for which the dense case performs relatively well? Testing around random performance might have some side effect.**
> > > >
> > > > We agree with you! We also adopt your advice to add some experiments. The corresponding results are given in **Appendix F** and **Figure 7 (Line 1026)**. Specifically, we illustrate performance trends across various pruning rates on the HellaSwag and BoolQ datasets. These results exhibit trends broadly consistent with those observed on CMMLU, further validating the stability of our proposed method.

---

> ### Comment · Reviewer_D7Lt · 2025-11-20
> **Reply to Rebuttal**
>
> I thank the authors for taking the time to carefully answer all the points I raised. I have revised their additions and modifications to the draft and I believe they have improved it. I will raise my score to accept.

---

> > ### Author Response · Authors · 2025-11-20
> > **Thank you!**
> >
> > Thank you for replying to our response and for raising score!

---

### Official Review · Reviewer_E6ZX · 2025-10-31

**Soundness:** 4
**Presentation:** 4
**Contribution:** 4
**Rating:** 8
**Confidence:** 3

**Summary:**

This paper presents a novel layer pruning method called Kernelized Dynamics Pruning (KDP). The method replaces several consecutive layers within a transformer model with a kernel mapping, linear transformation, and inverse mapping. This process is able to prune around 25% of the weights in the network while retaining higher accuracy than existing methods across several tasks. Theoretical results and comprehensive ablations explain the value of all components of the method and relate findings to previous observations in layer representational similarity.

**Strengths:**

There are several strengths that make this an interesting paper. The findings are quite well explained and strong. The findings hold consistently across a wide range of experimental settings. The theory relates findings to and builds upon previous results. Ablations demonstrate the value of each component. The method is explained clearly via figures, an algorithm, and descriptions.

**Weaknesses:**

N/A. This is a strong and well-organized paper.

**Questions:**

What is the cost of identifying and training the transformations? How does this compare to other layer pruning methods?

---

> ### Author Response · Authors · 2025-11-20
> **Rebuttal Reply 1/n**
>
> We would like to express our gratitude for your careful reading and valuable comments, which substantially improved our paper. In the following, we address your concerns one by one. Your comments are shown by **Q** (**Q** denotes Question) and are followed by our responses. For all page numbers and section numbers in this reply, please refer to the revised paper unless otherwise specified.
>
> >Q1: What is the cost of identifying and training the transformations? How does this compare to other layer pruning methods?
>
> **Layer Identification**: After extracting representations, this step only requires iterating over the similarities between representation layers as described in Algorithm 1. This process has a complexity of $O(n)$ and can be completed very rapidly.
>
> **Trianing**:  We have added these analysis supplementary materials to the **Appendix I**. As presented in **Section 3**, KDP introduces additional parameters during two steps (**Figure 2**): Step 1 (Section 3.1: Kernel Linearization Joint Training) requires training the parameters ($\theta$, {$\mathbf A_i$}) defined in **Equation (5)**; Step 2 (Section 3.2: Inverse Transformation Network) involves training the parameter $\phi$ from **Equation (6)**.
>
> Specifically, when the block of $K_{max}$  consecutive layers is replaced by KDP, the added parameter counts for the three components ($\theta$, {$\mathbf A_i$}, $\phi$) are as follows:
>
> 1. $\theta$: As detailed in Section 2.1, the RFF kernel learns a data-driven anisotropic Gaussian RBF kernel. Its learnable parameters $\theta$ are derived from the parameterization of the covariance matrix $\Sigma = \mathbf D + \mathbf{L}\mathbf{L}^{\top}$ (**Line 136**). This parameterization consists of two components: First,  the diagonal matrix $\mathbf D=\text{diag}(\exp(\lambda))$ (**Line 136**), is parameterized by the learnable vector $\lambda \in \mathbb{R}^d$ (**Line 137**), contributing $d$ parameters. Second, $\mathbf{L} \in \mathbb{R}^{d \times r}$ (**Line 137**) is a learnable low-rank factor matrix, which contributes $d \times r$ parameters. Therefore, the total parameter count is: $\text{Params}(\theta) = d + dr = d(1+r)$.
>
> 2. {$\mathbf A_i$}: Each operator $\mathbf A_i$ maps the kernel space representation $\varphi(\mathbf h_{l+i-1}) \in \mathbb{R}^{2m}$ to $\varphi(\mathbf h_{l+i}) \in \mathbb{R}^{2m}$. Consequently, $\mathbf A_i$ is a $(2m) \times (2m)$ matrix. To ensure stable training, $\mathbf A_i$ is parameterized as $\mathbf A_{i}=\mathbf I+\gamma_{i}\mathbf B_{i}$ (**Line 261**). The learnable parameters for each $\mathbf A_i$ thus consist of the scalar $\gamma_i$ and the matrix $\mathbf B_i \in \mathbb{R}^{2m \times 2m}$. This results in a total parameter count of: $\text{Params}(\\{\mathbf A_i\\}) = K_{max} \cdot (1 + (2m)^2)$. Notably, once training is complete, only the product $\left(\prod_{i=1}^k \widehat{\mathbf{A}}_i\right)$ (**Line 268**) needs to be stored. Consequently, during the inference phase, the final parameter count is reduced to only: $\text{Params}(\\{\mathbf A_i\\}) = (2m)^2$.
>
> 3. $\phi$: This  represents the parameters for the inverse transformation network $\mathcal{I}(\cdot)$, which is described in Section 3.2. The network is parameterized as $\mathcal{I}(x):=\alpha\cdot \text{MLP}(x)$ (**Line 293**), where $\alpha$ is a learnable scalar and $\text{MLP}(\cdot)$ stands for a two-layer MLP network. The total parameter count is given by: $\text{Params}(\phi) = 1 + (2m \cdot d_{\text{hidden}} + d_{\text{hidden}}) + (d_{\text{hidden}} \cdot d + d)$.
>
> Therefore, during the training phase, the total introduced parameter count is:
> $$\text{Params}^{\text{Trian}}(\text{KDP}) = d(1+r) + K_{max}(1 + (2m)^2) + (1 + d_{\text{hidden}}(2m + d + 1) + d),$$
> during the inference phase the total introduced parameter count is:
> $$\text{Params}^{\text{Inference}}(\text{KDP}) = d(1+r) + (2m)^2 + (1 + d_{\text{hidden}}(2m + d + 1) + d).$$
>
> Following the experimental setup described in the paper, we report the FLOPS and corresponding parameter counts for the two models in the table below.
>
> |LLM||FLOPs (G)||Params (M)||
> |:---|:---|:---|:---|:---|:---|
> |||**k=2**|**k=3**|**k=2**|**k=3**|
> |LLaMA-2-7B|Transformer Block|51.80|77.71|404.77|607.15|
> ||Fold Block|2.89||12.40||
> ||Retention Rate|5.59%|3.73%|3.07%|2.05%|
> |LLaMA-2-13B|Transformer Block|81.20|121.80|634.41|951.61|
> ||Fold Block|3.4||14.83||
> ||Retention Rate|4.19%|2.79%|2.34%|1.56%|
>
> Specifically, the "Transformer Block" row indicates the FLOPs or parameters of the corresponding $k$ consecutive layers. The "Fold Block" refers to the costs associated with the additional layers introduced by KDP. Finally, the "Retention Rate" is defined as the ratio of these extra parameters to the original parameters of the pruned layers.
> As shown in the table, the additional overhead introduced by our method is minimal.
> Finally, after pruning,  the LLaMA2-7B, LLaMA2-13B, and LLaMA3-8B models are reduced to 5.1B, 9.9B, and 6.0B parameters, respectively.

---

> > ### Author Response · Authors · 2025-11-20
> > **Rebuttal Reply 2/n**
> >
> > **Efficiency comparsion of other baselines**: We adopt your advice to add efficiency comparison experiments across different baselines, we have add this to **Table 10 (Line 1137)**. We present a computational efficiency comparison of different baseline models for Seq_len = 128, based on the settings from our Main Results (**Table 1**). The details are as follows:
> > | Model | Method | FLOPs (G) | Change (%) | SpeedUp |
> > | :--- | :--- | :--- | :--- | :--- |
> > | LLaMA-2-7B | Dense | 845.71 | - | - |
> > | | SLEB | 664.38 | 27.29 | 1.27x |
> > | | ShortGPT | 612.57 | 38.06 | 1.38x |
> > | | Streamline | 586.67 | 30.63 | 1.44x |
> > | | SliceGPT | 717.63 | 15.14 | 1.17x |
> > | | Ours | 650.06 | 23.13 | 1.30x |
> > | LLaMA-2-13B | Dense | 1645.01 | - | - |
> > | | SLEB | 1320.20 | 19.74 | 1.24x |
> > | | ShortGPT | 1239.00 | 24.68 | 1.33x |
> > | | Streamline | 1229.76 | 25.24 | 1.34x |
> > | | SliceGPT | 1392.89 | 15.32 | 1.18x |
> > | | Ours | 1255.99 | 23.64 | 1.30x |

---

### Official Review · Reviewer_d1gK · 2025-11-01

**Soundness:** 3
**Presentation:** 3
**Contribution:** 3
**Rating:** 6
**Confidence:** 3

**Summary:**

- The paper introduces Kernelized Dynamics Pruning (KDP), a layer-pruning method that interprets the forward pass of LLMs as a discrete-time dynamical system and leverages representational similarity to simplify internal dynamics.
- By projecting layer representations into a suitable kernel space, the method simplifies the non-linear transformations between consecutive layers and learns an inverse kernel transformation to reconstruct the pruned representations.
- The authors provide theoretical guarantees on the linearizability of representations in kernel space and empirical validation across 15 benchmarks, showing strong performance while maintaining efficiency and minimal retraining requirements.

**Strengths:**

- Interesting and well-motivated idea of replacing transformer blocks with lightweight modules that approximate their simple dynamics
- The paper establishes a theoretical error bound for kernel-space linearization, bridging empirical results with formal justification.
- KDP offers layer-wise compression while preserving performance better than other pruning methods.
- Extensive empirical validation as the method is evaluated on many benchmarks, suggesting robustness and generality of the proposed method.

**Weaknesses:**

- The paper primarily compares inference cost and FLOPs only against the original dense model (Table 10) rather than existing pruning baselines, limiting understanding of relative efficiency gains. It would provide a clearer and more fair comparison if the speedup was also compared across all the baselines
- The parameter count of the replacement modules are not clearly reported; it would be helpful if they were provided for the final pruned models that are evaluated
- While representation similarity is highlighted, further quantitative analysis of the “slow manifold” hypothesis or ablation on kernel choices would strengthen the conceptual link.

**Questions:**

- Is there any possible explanation for the significant rise in PPL for the Llama3.1-8B results? (in the case where there is no fine-tuning)
- Within this framework, is it possible or beneficial to fine-tune the modified model beyond the output projection layer? Specifically, could the pruned model, with its replacement modules, serve as a foundation for further task-specific fine-tuning, or are these modules primarily effective only when substituting already well-learned transformer representations?

---

> ### Author Response · Authors · 2025-11-20
> **Rebuttal Reply 1/n**
>
> We would like to express our gratitude for your careful reading and valuable comments, which substantially improved our paper. In the following, we address your concerns one by one. Your comments are shown by **W** and **Q** (**W** denotes Weakness; **Q** denotes Question) and are followed by our responses. For all page numbers and section numbers in this reply, please refer to the revised paper unless otherwise specified.
>
> >W1:  The paper primarily compares inference cost and FLOPs only against the original dense model (Table 10) rather than existing pruning baselines, limiting understanding of relative efficiency gains. It would provide a clearer and more fair comparison if the speedup was also compared across all the baselines.
>
> We agree with you!  And we adopt your advice to add efficiency comparison experiments across different baselines, we have add this to **Table 10 (Line 1137)**. The details are as follows.
>
> We present a computational efficiency comparison of different baseline models for Seq_len = 128, based on the settings from our Main Results (**Table 1**). The details are as follows:
> | Model | Method | FLOPs (G) | Change (%) | SpeedUp |
> | :--- | :--- | :--- | :--- | :--- |
> | LLaMA-2-7B | Dense | 845.71 | - | - |
> | | SLEB | 664.38 | 27.29 | 1.27x |
> | | ShortGPT | 612.57 | 38.06 | 1.38x |
> | | Streamline | 586.67 | 30.63 | 1.44x |
> | | SliceGPT | 717.63 | 15.14 | 1.17x |
> | | Ours | 650.06 | 23.13 | 1.30x |
> | LLaMA-2-13B | Dense | 1645.01 | - | - |
> | | SLEB | 1320.20 | 19.74 | 1.24x |
> | | ShortGPT | 1239.00 | 24.68 | 1.33x |
> | | Streamline | 1229.76 | 25.24 | 1.34x |
> | | SliceGPT | 1392.89 | 15.32 | 1.18x |
> | | Ours | 1255.99 | 23.64 | 1.30x |

---

> > ### Author Response · Authors · 2025-11-20
> > **Rebuttal Reply 2/n**
> >
> > >W2: The parameter count of the replacement modules are not clearly reported; it would be helpful if they were provided for the final pruned models that are evaluated.
> >
> > Thanks for your question. We have added these analysis supplementary materials to the **Appendix I**. As presented in **Section 3**, KDP introduces additional parameters during two steps(**Figure 2**): Step 1 (Section 3.1: Kernel Linearization Joint Training) requires training the parameters ($\theta$, {$\mathbf A_i$}) defined in **Equation (5)**; Step 2 (Section 3.2: Inverse Transformation Network) involves training the parameter $\phi$ from **Equation (6)**.
> >
> > Specifically, when we replace a block of $K_{max}$  consecutive layers, the parameter counts for the three components ($\theta$, {$\mathbf A_i$}, $\phi$) are as follows:
> > We have added these analysis supplementary materials to the **Appendix I**. As presented in **Section 3**, KDP introduces additional parameters during two steps (**Figure 2**): Step 1 (Section 3.1: Kernel Linearization Joint Training) requires training the parameters ($\theta$, {$\mathbf A_i$}) defined in **Equation (5)**; Step 2 (Section 3.2: Inverse Transformation Network) involves training the parameter $\phi$ from **Equation (6)**.
> >
> > Specifically, when the block of $K_{max}$  consecutive layers is replaced by KDP, the added parameter counts for the three components ($\theta$, {$\mathbf A_i$}, $\phi$) are as follows:
> >
> > 1. $\theta$: As detailed in Section 2.1, the RFF kernel learns a data-driven anisotropic Gaussian RBF kernel. Its learnable parameters $\theta$ are derived from the parameterization of the covariance matrix $\Sigma = \mathbf D + \mathbf{L}\mathbf{L}^{\top}$ (**Line 136**). This parameterization consists of two components: First,  the diagonal matrix $\mathbf D=\text{diag}(\exp(\lambda))$ (**Line 136**), is parameterized by the learnable vector $\lambda \in \mathbb{R}^d$ (**Line 137**), contributing $d$ parameters. Second, $\mathbf{L} \in \mathbb{R}^{d \times r}$ (**Line 137**) is a learnable low-rank factor matrix, which contributes $d \times r$ parameters. Therefore, the total parameter count is: $\text{Params}(\theta) = d + dr = d(1+r)$.
> >
> > 2. {$\mathbf A_i$}: Each operator $\mathbf A_i$ maps the kernel space representation $\varphi(\mathbf h_{l+i-1}) \in \mathbb{R}^{2m}$ to $\varphi(\mathbf h_{l+i}) \in \mathbb{R}^{2m}$. Consequently, $\mathbf A_i$ is a $(2m) \times (2m)$ matrix. To ensure stable training, $\mathbf A_i$ is parameterized as $\mathbf A_{i}=\mathbf I+\gamma_{i}\mathbf B_{i}$ (**Line 261**). The learnable parameters for each $\mathbf A_i$ thus consist of the scalar $\gamma_i$ and the matrix $\mathbf B_i \in \mathbb{R}^{2m \times 2m}$. This results in a total parameter count of: $\text{Params}(\\{\mathbf A_i\\}) = K_{max} \cdot (1 + (2m)^2)$. Notably, once training is complete, only the product $\left(\prod_{i=1}^k \widehat{\mathbf{A}}_i\right)$ (**Line 268**) needs to be stored. Consequently, during the inference phase, the final parameter count is reduced to only: $\text{Params}(\\{\mathbf A_i\\}) = (2m)^2$.
> >
> > 3. $\phi$: This  represents the parameters for the inverse transformation network $\mathcal{I}(\cdot)$, which is described in Section 3.2. The network is parameterized as $\mathcal{I}(x):=\alpha\cdot \text{MLP}(x)$ (**Line 293**), where $\alpha$ is a learnable scalar and $\text{MLP}(\cdot)$ stands for a two-layer MLP network. The total parameter count is given by: $\text{Params}(\phi) = 1 + (2m \cdot d_{\text{hidden}} + d_{\text{hidden}}) + (d_{\text{hidden}} \cdot d + d)$.
> >
> > Therefore, during the training phase, the total introduced parameter count is:
> > $$\text{Params}^{\text{Trian}}(\text{KDP}) = d(1+r) + K_{max}(1 + (2m)^2) + (1 + d_{\text{hidden}}(2m + d + 1) + d),$$
> > during the inference phase the total introduced parameter count is:
> > $$\text{Params}^{\text{Inference}}(\text{KDP}) = d(1+r) + (2m)^2 + (1 + d_{\text{hidden}}(2m + d + 1) + d).$$
> >
> > Following the experimental setup described in the paper, we report the FLOPS and corresponding parameter counts for the two models in the table below.
> >
> > |LLM||FLOPs (G)||Params (M)||
> > |:---|:---|:---|:---|:---|:---|
> > |||**k=2**|**k=3**|**k=2**|**k=3**|
> > |LLaMA-2-7B|Transformer Block|51.80|77.71|404.77|607.15|
> > ||Fold Block|2.89||12.40||
> > ||Retention Rate|5.59%|3.73%|3.07%|2.05%|
> > |LLaMA-2-13B|Transformer Block|81.20|121.80|634.41|951.61|
> > ||Fold Block|3.4||14.83||
> > ||Retention Rate|4.19%|2.79%|2.34%|1.56%|
> >
> > Specifically, the "Transformer Block" row indicates the FLOPs or parameters of the corresponding $k$ consecutive layers. The "Fold Block" refers to the costs associated with the additional layers introduced by KDP. Finally, the "Retention Rate" is defined as the ratio of these extra parameters to the original parameters of the pruned layers.
> >
> > Finally, after pruning,  the LLaMA2-7B, LLaMA2-13B, and LLaMA3-8B models are reduced to 5.1B, 9.9B, and 6.0B parameters, respectively.

---

> > > ### Author Response · Authors · 2025-11-20
> > > **Rebuttal Reply 3/n**
> > >
> > > >W3: While representation similarity is highlighted, further quantitative analysis of the "slow manifold" hypothesis or ablation on kernel choices would strengthen the conceptual link.
> > >
> > > Thank you for this insightful comment. We appreciate the suggestion and have added a more detailed analysis of the slow manifold in **Appendix H**.
> > >
> > > **1.On further analysis of the slow manifold hypothesis:**
> > > First, we provide further, detailed explanations of our perspective on the slow manifold hypothesis. Our method is inspired by the well-known "manifold hypothesis," which posits that high-dimensional data often reside on a manifold with a dimensionality much lower than that of the ambient space. This hypothesis has also proven beneficial in various fields, including diffusion models [1].
> > >
> > > This leads us to consider whether this property also holds for the model's internal representations, allowing them to be embedded in a low-dimensional manifold.  The internal representations of the model are sequential. As shown in **Equation 1**, in standard Pre-Norm Transformers, $\mathbf h_{l+1}$ is obtained by adding an increment to $\mathbf h_l$. We therefore define this increment as the velocity of change in the representation.
> > > Although CKA and Cosine similarity are measures of similarity rather than distance metrics, they are closely correlated with distance measures on the manifold. This implies that high similarity between two representations corresponds to a small distance between them.  The Cosine and CKA similarity matrices (**Figure 1**) provide quantitative evidence for this. The extensive bright red regions within these matrices, particularly for the CKA measure, indicate that the representations across multiple consecutive layers are highly similar, with scores approaching 1.0.  When the representation change over a single propagation step is small, we consider the representation to be in a stable state on the manifold. It is then often feasible to model this stable evolution using a simple dynamical model.
> > >
> > > The concept of using linear approximations for nonlinear dynamics finds parallels in many fields. Beyond the fundamental example of calculus, a representative example is Linear Stability Analysis, which is widely applied in partial differential equations (PDE).
> > >
> > > To provide further intuitive support for our hypothesis, we visualize the representational trajectories across different layers for 100 samples from LLaMA2-7B using t-SNE (**Figure 10**). The resulting plot clearly exhibits a strong linear trend, which in turn validates the feasibility of our method.
> > >
> > > **2.On the Ablation of Kernel Functions:**
> > > The choice of kernel function is a key aspect of this method. We first address a fundamental question: Is the kernel function necessary? As shown in **Table 3**, performing linear fitting in the original space (the OLS method) leads to a catastrophic performance collapse. This quantitatively demonstrates that the layer-to-layer transformation in Transformers is highly non-linear. Consequently, any attempt at linear simplification in the original space results in severe information loss. Therefore, using the "kernel trick" to map representations into an linear space is a necessary step for simplifying the dynamics.
> > > Furthermore, we focus on a learnable RFF kernel  whose universal approximation capability is verified by **Lemma 1**.  We leave the exploration of additional kernel constructions for future work.
> > >
> > > [1] Song Y, Ermon S. Generative modeling by estimating gradients of the data distribution[J]. Advances in neural information processing systems, 2019, 32.

---

> > > > ### Author Response · Authors · 2025-11-20
> > > > **Rebuttal Reply 4/n**
> > > >
> > > > >Q1: Is there any possible explanation for the significant rise in PPL for the Llama3.1-8B results? (in the case where there is no fine-tuning)
> > > >
> > > > Thank you for your question.
> > > > We think that the significant PPL increase of KDP on LLaMA-3-8B may be attributed to the model's greater internal homogeneity.
> > > > As shown in **Figure 1**, the CKA matrix for LLaMA-2 (7B/13B) exhibits a distinct "blocky" structure, where high similarity (red) is primarily concentrated in specific regions near the diagonal.
> > > > The CKA matrix of LLaMA-3-8B, in contrast, forms a nearly solid, homogeneous red square, indicating that almost all layers in its network are highly similar to one another.
> > > > **Theorem 1** reveals that approximation errors accumulate. Although the theorem analyzes the error within a single k-step block, this principle of accumulation applies equally to the entire network.
> > > > In LLaMA-3-8B, due to its homogeneous structure, the function of each layer is highly coupled with those of other layers, when we replace the first block (e.g., [6, 7]), KDP introduces a small approximation error, $\mathcal{E}_1$, when we subsequently replace a second block (e.g., [9, 10]), its input (from layer 8) has already been affected by $\mathcal{E}_1$. Therefore, the second error introduced by KDP, $\mathcal{E}_2$, builds upon $\mathcal{E}_1$.
> > > > In LLaMA-3-8B, as the pruning rate increases (i.e., more blocks are replaced), the total approximation error compounds non-linearly throughout the network. The sharp surge in PPL (Figure 6, right) occurs precisely when the total pruning rate exceeds a critical threshold, at which point this global, cumulative error overwhelms the model's stability.
> > > >
> > > > >Q2: Within this framework, is it possible or beneficial to fine-tune the modified model beyond the output projection layer? Specifically, could the pruned model, with its replacement modules, serve as a foundation for further task-specific fine-tuning, or are these modules primarily effective only when substituting already well-learned transformer representations?
> > > >
> > > > We greatly appreciate you for raising this important question regarding fine-tuning flexibility. We conduct dedicated experiments to investigate this point, and the results confirm the latter part of the your conjecture: The core utility of the KDP module lies in its ability to efficiently replace well-trained Transformer representations, rather than serving as a fine-tunable base module for learning new tasks.
> > > > We employ the LLaMA-2-7B model and the "Standard" calibration dataset (4000 samples) describe in **Table 11 (Line 1168)**. As shown in the table, fine-tuning the KDP module itself significantly degrad model performance, causing it to fall even below that of the standard KDP without any post-training. This result strongly supports our core contribution (i.e., **Contribution 1 (Line 77)**): KDP is not intended to construct a smaller, trainable sub-network, but rather aims to 'find a geometric perspective that reveals the intrinsic simplicity of complex dynamics.'
> > > > | Model | Method | CMNLI | HeSW | PIQA | CHID | WSC | CoQA | BoolQ | MMLU | CMMLU | Race | SST2 | C3 | Avg |
> > > > | :--- | :--- | :--- | :--- | :--- | :--- | :--- | :--- | :--- | :--- | :--- | :--- | :--- | :--- | :--- |
> > > > | LLaMA-2-7B | KDP | 33.6 | 65.1 | 70.1 | 27.2 | 73.9 | 45.4 | 71.6 | 44.7 | 28.1 | 39.9 | 94.0 | 43.7 | 53.11 |
> > > > | | KDP (Post Train) | 31.7 | 47.2 | 63.1 | 25.3 | 71.0 | 33.4 | 70.3 | 38.5 | 24.8 | 40.0 | 88.6 | 40.3 | 47.85 |
> > > >
> > > > Specifically, the primary reason for this phenomenon lies in KDP's distinct training supervision. The KDP module is trained using local representations as the supervision signal. Its entire optimization objective is to learn a geometric mapping ($\varphi_\theta$) and an inverse mapping ($\mathcal{I}_\phi$) in order to approximate the original, complex non-linear representation flow with a simple linear transformation ${\mathbf{A}_i}$ in the kernel space.
> > > >
> > > > Second, KDP possesses limited expressive power, KDP module we design is far from comparable, in terms of both parameter count and expressivity, to the entire multi-layer Transformer block (including MHA and FFN) it replaces. When we subject this KDP module to task fine-tuning, we are essentially forcing this "simplified module" to play the role of a "full-function Transformer module." The module's architecture is not designed to learn complex downstream task knowledge. Consequently, fine-tuning instead disrupts the geometric structure it has already acquired for precise representation approximation, resulting in performance degradation.
> > > >
> > > > In summary, our KDP module is designed to function as an efficient "representation flow simplifier" and is highly effective at replacing well-trained representations. Meanwhile, the model's task-adaptation fine-tuning should be delegated to other, more expressive components within the network (such as the output layer fine-tuning detailed in  **Appendix B**).

---

### Official Review · Reviewer_1BU2 · 2025-11-12

**Soundness:** 4
**Presentation:** 4
**Contribution:** 4
**Rating:** 6
**Confidence:** 5

**Summary:**

The paper introduces a novel layer pruning method for LLMs called KDP. The key idea is to view a forward pass as a discrete-time dynamical system, where high similarity between consecutive layers implies entry into a slow manifold, indicating redundant computation. KDP projects intermediate representations into a Hilbert space, where complex nonlinear dynamics between layers become approximately linear. It then learns linear operators to approximate transitions and an inverse mapping network to reconstruct the representations back in the original space, enabling whole-layer pruning without retraining. Experiments across benchmarks indicate that KDP is a performant structured pruning method on LLMs.

**Strengths:**

The method is novel, significantly different from previous pruning methods.

The evaluation is conducted on various models of different scales; benchmarks are comprehensive.

The paper is overall easy to follow.

**Weaknesses:**

My major concern lies in the additional overhead introduced by KDP. Kernel mapping and linear operators might introduce non-trivial costs.

**Questions:**

What is the additional overhead of KDP compared to the LLM? Providing time or FLOPs analysis might be helpful.

---

> ### Author Response · Authors · 2025-11-20
> **Rebuttal Reply 1/n**
>
> We would like to express our gratitude for your careful reading and valuable comments, which substantially improved our paper. In the following, we address your concerns one by one. Your comments are shown by **W** and **Q** (**W** denotes Weakness; **Q** denotes Question) and are followed by our responses. For all page numbers and section numbers in this reply, please refer to the revised paper unless otherwise specified.
>
> >W1 and Q1: My major concern lies in the additional overhead introduced by KDP. Kernel mapping and linear operators might introduce non-trivial costs. What is the additional overhead of KDP compared to the LLM? Providing time or FLOPs analysis might be helpful.
>
> Thanks for your question. We have added these analysis supplementary materials to the **Appendix I**. As presented in **Section 3**, KDP introduces additional parameters during two steps (**Figure 2**): Step 1 (Section 3.1: Kernel Linearization Joint Training) requires training the parameters ($\theta$, {$\mathbf A_i$}) defined in **Equation (5)**; Step 2 (Section 3.2: Inverse Transformation Network) involves training the parameter $\phi$ from **Equation (6)**.
> +
> Specifically, when the block of $K_{max}$  consecutive layers is replaced by KDP, the added parameter counts for the three components ($\theta$, {$\mathbf A_i$}, $\phi$) are as follows:
>
> 1. $\theta$: As detailed in Section 2.1, the RFF kernel learns a data-driven anisotropic Gaussian RBF kernel. Its learnable parameters $\theta$ are derived from the parameterization of the covariance matrix $\Sigma = \mathbf D + \mathbf{L}\mathbf{L}^{\top}$ (**Line 136**). This parameterization consists of two components: First,  the diagonal matrix $\mathbf D=\text{diag}(\exp(\lambda))$ (**Line 136**), is parameterized by the learnable vector $\lambda \in \mathbb{R}^d$ (**Line 137**), contributing $d$ parameters. Second, $\mathbf{L} \in \mathbb{R}^{d \times r}$ (**Line 137**) is a learnable low-rank factor matrix, which contributes $d \times r$ parameters. Therefore, the total parameter count is: $\text{Params}(\theta) = d + dr = d(1+r)$.
>
> 2. {$\mathbf A_i$}: Each operator $\mathbf A_i$ maps the kernel space representation $\varphi(\mathbf h_{l+i-1}) \in \mathbb{R}^{2m}$ to $\varphi(\mathbf h_{l+i}) \in \mathbb{R}^{2m}$. Consequently, $\mathbf A_i$ is a $(2m) \times (2m)$ matrix. To ensure stable training, $\mathbf A_i$ is parameterized as $\mathbf A_{i}=\mathbf I+\gamma_{i}\mathbf B_{i}$ (**Line 261**). The learnable parameters for each $\mathbf A_i$ thus consist of the scalar $\gamma_i$ and the matrix $\mathbf B_i \in \mathbb{R}^{2m \times 2m}$. This results in a total parameter count of: $\text{Params}(\\{\mathbf A_i\\}) = K_{max} \cdot (1 + (2m)^2)$. Notably, once training is complete, only the product $\left(\prod_{i=1}^k \widehat{\mathbf{A}}_i\right)$ (**Line 268**) needs to be stored. Consequently, during the inference phase, the final parameter count is reduced to only: $\text{Params}(\\{\mathbf A_i\\}) = (2m)^2$.
>
> 3. $\phi$: This  represents the parameters for the inverse transformation network $\mathcal{I}(\cdot)$, which is described in Section 3.2. The network is parameterized as $\mathcal{I}(x):=\alpha\cdot \text{MLP}(x)$ (**Line 293**), where $\alpha$ is a learnable scalar and $\text{MLP}(\cdot)$ stands for a two-layer MLP network. The total parameter count is given by: $\text{Params}(\phi) = 1 + (2m \cdot d_{\text{hidden}} + d_{\text{hidden}}) + (d_{\text{hidden}} \cdot d + d)$.
>
> Therefore, during the training phase, the total introduced parameter count is:
> $$\text{Params}^{\text{Trian}}(\text{KDP}) = d(1+r) + K_{max}(1 + (2m)^2) + (1 + d_{\text{hidden}}(2m + d + 1) + d),$$
> during the inference phase the total introduced parameter count is:
> $$\text{Params}^{\text{Inference}}(\text{KDP}) = d(1+r) + (2m)^2 + (1 + d_{\text{hidden}}(2m + d + 1) + d).$$
>
> Following the experimental setup described in the paper, we report the FLOPS and corresponding parameter counts for the two models in the table below.
>
> |LLM||FLOPs (G)||Params (M)||
> |:---|:---|:---|:---|:---|:---|
> |||**k=2**|**k=3**|**k=2**|**k=3**|
> |LLaMA-2-7B|Transformer Block|51.80|77.71|404.77|607.15|
> ||Fold Block|2.89||12.40||
> ||Retention Rate|5.59%|3.73%|3.07%|2.05%|
> |LLaMA-2-13B|Transformer Block|81.20|121.80|634.41|951.61|
> ||Fold Block|3.4||14.83||
> ||Retention Rate|4.19%|2.79%|2.34%|1.56%|
>
> Specifically, the "Transformer Block" row indicates the FLOPs or parameters of the corresponding $k$ consecutive layers. The "Fold Block" refers to the costs associated with the additional layers introduced by KDP. Finally, the "Retention Rate" is defined as the ratio of these extra parameters to the original parameters of the pruned layers.
> As shown in the table, the additional overhead introduced by our method is minimal.

---

> > ### Comment · Reviewer_1BU2 · 2025-11-20
> > **Concerns Resolved**
> >
> > According the authors' response, I believe that the extra overhead introduced by KDP is small compared with the cost of Transformer blocks. My major overhead concerns are resolved, and I will raise my scores.

---

> > > ### Author Response · Authors · 2025-11-20
> > > **Thank you!**
> > >
> > > Thank you for replying to our response and for raising score!

---

### Author Response · Authors · 2025-11-20
**General Response**

We thank the reviewers for their time, effort, and recognition of our paper. In response to their comments and our own further scrutiny, we have optimized and updated the following content in the new version:
1. Correct typos, refine phrasing in the main text, and update key citations;
2. Introduce fundamental concepts regarding generalization theory (see Appendix A);
3. Validate performance sensitivity to varying pruning rates on the HellaSwag and BoolQ datasets (see Figure 7 in Appendix F);
4. Include an efficiency analysis of the baselines (see Table 10 in Appendix F);
5. Provide further analysis and experiments concerning the slow manifold (see Figure 10 in Appendix H);
6. Conduct additional analysis and experiments on computational overhead (see Appendix I).
We have highlighted the key additions and revisions in blue. We are happy to address any further comments or questions during the remaining days of the discussion period.

---

### Author Response · Authors · 2025-12-02
**Summary of Rebuttal Status & Guide to Revisions**

**To the newly assigned Area Chair:**

We understand the significant workload you are facing. To assist you, we provide a concise summary of the reviewer consensus reached before the system freeze, and a pointer to our detailed revision log.

**1. Current Status & Consensus:** Following the rebuttal interactions, our submission achieved an average score of **7.5 (Scores: 8, 8, 8, 6)**, with all reviewers leaning towards acceptance.

The score increase (from Reviewer ``1BU2`` and ``D7Lt``) and the consensus were finalized on **20, Nov, 2025**. This precedes the public disclosure of the OpenReview bug (Nov 27), ensuring these evaluations reflect genuine scientific consensus unaffected by recent irregularities.

● **Reviewer ``1BU2``**: Increased score from 6 to 8. Explicitly stated: "*My major overhead concerns are resolved, and I will raise my scores.*"

● **Reviewer  ``d1gK``**: Rated 6 (Positive). We provided detailed responses covering all their questions.

● **Reviewer  ``E6ZX``**: Rated 8 (Positive). We provided detailed responses covering all their questions.

● **Reviewer  ``D7Lt``**: Increased score from 6 to 8. Explicitly stated: "*I have revised their additions and modifications to the draft and I believe they have improved it. I will raise my score to accept.*"

---

**2. Summary of Revisions:** To respect your time and avoid redundancy, we will not repeat the full details here. Please refer to our **"General Response"** (posted on **20, Nov, 2025**) for the comprehensive list of changes.
We hope this summary helps you navigate the review history efficiently.

Sincerely, The Authors

---

### Meta-Review · Area_Chair_KGAG · 2026-01-04

**Summary:**

The paper received positive ratings from all reviewers, with no major concerns raised after the rebuttal. Therefore, it is recommended for acceptance.

**Reviewer Concerns:**

No major concerns remain following the rebuttal.

**Reviewer Scores:**

Some reviewers may increase their scores after the rebuttal.

---

### Decision · Program_Chairs · 2026-01-26

Accept (Poster)